# Persistence of moist plumes from overshooting convection in the Asian monsoon anticyclone

Sergey M. Khaykin[1], Elizabeth Moyer[2], Martina Krämer[3], Benjamin Clouser[2], Silvia Bucci[4,a], Bernard Legras[4], Alexey Lykov[5], Armin Afchine[3], Francesco Cairo[6], Ivan Formanyuk[5], Valentin Mitev[7], Renaud Matthey[8], Christian Rolf[3], Clare Singer[2,b], Nicole Spelten[3], Vasiliy Volkov[5], Vladimir Yushkov[5] and Fred Stroh[3]

[1] Laboratoire Atmosphères, Observations Spatiales (LATMOS), UVSQ, Sorbonne Université, CNRS, IPSL, Guyancourt, France
[2] Dept. of the Geophysical Sciences, University of Chicago, Chicago, IL, USA
[3] Forschungszentrum Jülich, Institut für Energie und Klimaforschung (IEK-7), Germany
[4] Laboratoire de Météorologie Dynamique (LMD), CNRS, IPSL, ENS-PSL, École Polytechnique, Sorbonne Université, Paris, France
[5] Central Aerological Observatory of RosHydroMet, Dolgoprudny, Russian Federation
[6] National Research Council of Italy, Institute of Atmospheric Sciences and Climate (CNR-ISAC), Rome, Italy
[7] Centre Suisse d'Electronique et de Microtechnique Neuchâtel, Switzerland.
[8] University of Neuchâtel, Neuchâtel, Switzerland
[a] Now at Institut für Meteorologie und Geophysik, University of Vienna, Vienna, Austria
[b] Now at Dept. of Environmental Science and Engineering, California Institute of Technology, Pasadena, CA, USA

*Correspondence to*: Sergey M. Khaykin (sergey.khaykin@latmos.ipsl.fr)

**Abstract.** The Asian Monsoon Anticyclone (AMA) represents one of the wettest regions in the lower stratosphere (LS) and is a key contributor to the global annual maximum in LS water vapour. While the AMA wet pool is linked with persistent convection in the region and horizontal confinement of the anticyclone, there remain ambiguities regarding the role of tropopause-overshooting convection in maintaining the regional LS water vapour maximum. This study tackles this issue using a unique set of observations from onboard the high-altitude M55-Geophysica aircraft deployed in Nepal in Summer 2017 within the EU StratoClim project. We use a combination of airborne measurements (water vapour, ice water, water isotopes, cloud backscatter) together with ensemble trajectory modeling coupled with satellite observations to characterize the processes controlling water vapour and clouds in the confined lower stratosphere (CLS) of AMA. Our analysis puts in evidence the dual role of overshooting convection, which may lead to hydration or dehydration depending on the synoptic-scale tropopause temperatures in AMA. We show that all of the observed CLS water vapour enhancements are traceable to convective events within AMA and furthermore bear an isotopic signature of the overshooting process. A surprising result is that the plumes of moist air with mixing ratios nearly twice the background level can persist for weeks whilst recirculating within the anticyclone, without being subject to irreversible dehydration through ice settling. Our findings highlight the importance of convection and recirculation within AMA for the transport of water into the stratosphere.

## 1 Introduction

Water vapour in the lower stratosphere has a direct impact on surface climate and stratospheric ozone chemistry (e.g. Dessler et al., 2013, Dvortsov and Solomon, 2001; Anderson et al., 2017). The variability of global lower stratospheric water vapour is, to first order, regulated by the minimum temperature in the Tropical Tropopause Layer (TTL) – the main

gateway for stratospheric entry of tropospheric moisture (e.g., Fueglistaler et al., 2009, and references therein). This way, the variation of stratospheric water vapour follows the annual cycle of TTL temperatures experiencing the minimum during Austral summer, when the TTL is coldest and driest above the western Pacific region (e.g. Randel and Jensen, 2013). During Boreal summer, one of the primary contributor to the annual maximum of lower stratospheric water vapour is the Asian monsoon (e.g. Bannister et al., 2004; Fueglistaler et al., 2005; Ueyama et al., 2018) and the associated Asian summer monsoon anticyclone (AMA).

The AMA is one of the largest atmospheric circulation features on Earth, owing its existence to frequent deep convection above Southern Asia and the Bay of Bengal, strong surface heating over the Tibetan plateau and orographic updrafts at the Southern slopes of Himalayas (e.g., Hoskins and Rodwell, 1995). The AMA is characterized by a persistent maximum of water vapour extending up to 68 hPa level (Park et al., 2007; Santee et al., 2017), which makes it the wettest region in the Boreal summer lower stratosphere. Generally, this large-scale maximum is conditioned by convective uplift of moist air in the Asian monsoon region and its horizontal confinement within the anticyclone (Dethof et al., 1999; Ploeger et al., 2015).

The transport of moist air into the stratosphere occurs via different pathways: slow radiatively-driven ascent (e.g. Garny and Randel, 2016), fast convective overshooting (e.g. Fu et al., 2006) and adiabatic transport across the horizontal boundaries of AMA (e.g. Pan et al., 2016) into the tropical and midlatitude stratosphere (Randel et al., 2010; Wright et al., 2011; Dethof et al., 1999; Vogel et al., 2016; Rolf et al., 2018; Nutzel et al., 2019). The role of different transport pathways, particularly of the convective overshooting and its predominant source regions, is subject of ongoing debate.

A relatively small impact of overshooting convection on AMA humidity is found by James et al. (2008), Wright et al. (2011), Randel et al. (2015) and Zhang et al. (2016), whereas Fu et al. (2006), Ueyama et al. (2018) and Brunamonti et al. (2018) suggest that this process can be an important contributor to the total water in the Asian TTL. The convective impact of different source regions (e.g., Bay of Bengal, Tibetan Plateau, Southern slopes of Himalayas or Sichuan basin) is also under debate (Bannister et al., 2004; Bergman et al., 2012; Fu et al., 2006; Lelieveld et al., 2007; Wright et al., 2011; Devasthale & Fueglistaler, 2010; James et al., 2008; Park et al., 2007; Tissier & Legras, 2016; Legras & Bucci, 2020). In general, there is no consensus regarding the primary convective source regions, nor regarding the net convective effect of deep convection on the CLS water vapour, which points out the complexity of physical processes in the AMA system.

Notably, most of the observational evidence regarding the mechanisms controlling AMA water is derived from passive satellite measurements, which due to their coarse vertical resolution cannot resolve the small-scale processes such as moistening produced by localized injections of ice. The high-resolution measurements of water vapour within AMA using small balloons and research aircrafts have only recently become available (Bian et al., 2012; Gottschaldt et al., 2018; Vernier et al., 2018; Brunamonti et al., 2018). The most extensive set of high-resolution measurements including water vapour, ice water, water isotopic ratio, aerosols and various tracers is provided by the StratoClim aircraft campaign, which was held in July-August 2017 and involved the high-altitude M55-Geophysica aircraft deployed in Kathmandu, Nepal (Stroh et al., 2021, in prep. same issue; Kṛämer et al., 2020).

In this study, we combine local airborne measurements with global satellite observations to characterize the mechanisms of convective impact on water vapour and clouds through both mass and energy transport above the cold point tropopause. We provide observational evidence of convectively-induced lower stratosphere hydration and dehydration of both irreversible and reversible types. The link between the local variations and phase transitions of water with deep convection across the Asian anticyclone is investigated using ensemble trajectory modeling constrained by

satellite detections of convective cloud tops. Section 2 of this article describes the experimental and modeling setup, Section 3 provides the satellite view of synoptic-scale development in the TTL during the campaign period and presents the ensemble of airborne measurements. The convective source regions and their AMA-wide effects on water vapour are analyzed in Sect. 4. Section 5 documents and analyzes the observed processes controlling water vapour above the tropopause, and is followed by the discussion and summary in Sect. 6.

## 2      Data and methods

### 2.1   StratoClim campaign and airborne instruments

The main experiment of the EU Framework Programme 7 (FP7) StratoClim project was the deployment of the Russian high-altitude M55-Geophysica aircraft in Kathmandu, Nepal during July-August 2017. The campaign included eight flights (hereinafter referred to as F$x$, where $x$ is the flight number) performed every second day during the period 27 July – 10 August in both the morning and afternoon hours. Three of the flights were performed within the Nepali borders, whereas in the rest of the flights the airplane flew out to southwest, south and southeast from Nepal reaching the Bay of Bengal (see Fig. S1 of the Supplement). The Geophysica aircraft hosted a large number of in situ and remote sensors for measuring gaseous and particulate UTLS composition. A full description of the campaign is provided by Stroh et al. (2021, in prep. same issue). In this study, we use in situ measurements of water vapour, total water and water isotopologues respectively by FLASH, FISH and ChiWIS instruments as well as particle backscatter measurements by the onboard MAS scatterometer and MAL lidar.

### 2.1.1 In situ water measurements

FLASH-A (Fluorescent Lyman-Alpha Stratospheric Hygrometer for Aircraft) is an airborne instrument of the FLASH hygrometer family designed specifically for the M55-Geophysica aircraft (Sitnikov et al, 2007). The instrument was redesigned in 2009 (Khaykin et al., 2013) for the RECONCILE campaign (von Hobe et al., 2013) and substantially improved for the StratoClim experiment. FLASH-A is mounted inside a gondola under the right wing of Geophysica and has a rear-facing inlet, enabling water vapour measurements. With the aspiration rate of 470 cm$^3$/s, the air samples in a 90 cm$^3$ measurement chamber are fully exchanged every 0.19 s. The chamber is maintained at constant temperature (24 °C) and pressure (36 hPa). Before a flight, the instrument is ventilated for several hours using dry air (< 1 ppmv) whereas the inlet tube, heated to 30 °C, is kept sealed before the aircraft climbs to 250 hPa level to avoid chamber contamination by moist tropospheric air.

Unlike the previous airborne versions of FLASH-A with transverse optical setup, the StratoClim FLASH-A rendition has a coaxial optics similar to that of the FLASH-B balloon-borne instrument (Yushkov et al., 1998). The water vapour mixing ratio is detected by sensing the fluorescence light yielded by photodissociation of water molecules after their exposure to Lyman-alpha radiation. A near Lyman-$\alpha$ line (123.6 nm), is produced by a krypton lamp whereas the hydroxyl fluorescence at 300 – 325 nm wavelength range is detected by a photomultiplier operating in photon-counting mode. The accuracy of water vapour measurements in 1 – 100 ppmv range is estimated at 8%, whereas the precision of 1 Hz data in the stratosphere is 0.2 ppmv with a detection limit of 0.1 ppmv for 5 s integration time. FLASH-A was calibrated against a reference MBW-373L frost-point hygrometer before and after the aircraft deployment as well as during the campaign

using FISH calibration facility. During StratoClim campaign, FLASH-A operated in all the eight scientific flights as well as during the transfer flight to Kathmandu.

ChiWIS (Chicago Water Isotope Spectrometer) is an airborne implementation of the ChiWIS-lab instrument (Sarkozy et al., 2020) designed for atmospheric chamber measurements of water vapour and water isotopologues under UTLS conditions, i.e. low temperature and humidity environment. The new version of the instrument is a tunable diode laser (TDL), off-axis integrated cavity output spectrometer (Clouser et al., 2021, in prep. same issue). The spectrometer scans absorption lines of both $H_2O$ and HDO near 2.647 µm wavelength in a single current sweep. With a 90 cm-long multi-pass cell, the effective path length amounts to more than 7 km. During the airborne campaign, the instrument has demonstrated measurement precision for ten second integration times of 18 ppbv and 80 pptv in $H_2O$ and HDO, respectively. The measurements were reported at 0.2 – 0.5 Hz frequency depending on the ambient mixing ratio and the desired signal-to-noise ratio. Periods of the flights where the internal cell pressure of ChiWIS was below 30 hPa are not reported because of the large influence of vapour desorption from the cavity walls. ChiWIS reported measurements for all the StratoClim flights except F1 and F5.

FISH (Fast In situ Stratospheric Hygrometer) is a closed-path Lyman-α fluorescence hygrometer with a forward-facing inlet, which enables measurement of total water (sum of gas phase water and sublimated ice crystals). The measurement accuracy is 6 % – 8 % whereas the precision of 1 Hz data is estimated at 0.3 ppmv (Zöger et al., 1999; Meyer et al., 2015). Inside the cirrus clouds, the ice water content (IWC) is calculated by subtracting the FLASH-A gas-phase water from the total water measured by FISH, as described by Afchine et al. (2018). The minimum detectable IWC is $3 \times 10^{-2}$ ppmv ($\sim 3 \times 10^{-3}$ mg m$^{-3}$). The FISH instrument has provided measurements for flights F2, F4, F6, F7 and F8.

The point-by-point intercomparison between FLASH-A, ChiWIS and FISH clear-air measurements reported by Singer et al. (2021, in prep. same issue) revealed a remarkable degree of agreement and an equally-high capacity of all hygrometers to resolve fine-scale spatial structures in UTLS water vapour. In clear-sky periods at mixing ratios below 10 ppmv, the mean bias between FISH and FLASH-A was -1.47% with an $r^2$ value of 0.930. For ChiWIS and FLASH-A, the mean bias was -1.42% and +0.74% with $r^2$ values of 0.928 and 0.930 for clear-sky and in-cloud periods at mixing ratios below 10 ppmv, respectively. Singer et al. (2021, in prep. same issue)also found good agreement of the airborne measurements with the collocated MLS water vapour profiles as well as with concomitant balloon soundings in Dhulikhel, Nepal (Brunamonti et al., 2018) using a cryogenic frost point hygrometer (CFH) instrument. Altogether, this provides a high degree of confidence in the StratoClim water vapour measurement.

### 2.1.2 In situ temperature and cloud measurements

The temperature was measured by TDC (ThermoDynamic Complex), a modified Rosemount 5-hole probe that provides an accuracy of 0.5 K and precision of 0.1 K for temperature measurements at 1 Hz frequency (Shur et al., 2007). We used TDC measurements of temperature and pressure to convert FLASH-A water vapour mixing ratio into relative humidity over ice (RHi) as well as to compute the saturation mixing ratio using the saturation vapour pressure equation by Murphy and Koop (2005). The accuracy of TDC measurements is discussed by Singer et al. and Stroh et al. (2021, in prep, same issue)

NIXE-CAPS (New Ice eXpEriment: Cloud and Aerosol Particle Spectrometer) is mounted under the right wing of Geophysica and measures the cloud particle number size distributions in the size range of 3–930 µm diameter at a time resolution of 1 Hz (Meyer, 2012). The IWC derived from particle size distributions are found to be in good agreement

with those derived from FISH total water measurements (Afchine et al., 2018). The lower detection limit of the instrument is 0.05 ppmv (≈0.005 mg m$^{-3}$). NIXE-CAPS provided measurements in all the flights.

In situ measurements of cloud/aerosol backscatter and with a time constant of 10 s were provided by the forward-looking backscatter probe MAS (Multiwavelength Aerosol Scattersonde) described by Buontempo et al. (2006). To distinguish between clear-sky and in-cloud measurements, here we use a threshold of 1.2 units of backscatter ratio at 532 nm together with a 2.5% threshold in the volume depolarization (corresponding to particle depolarization of 8-10%). MAS instruments operated in all the flights except F1, for which we used NIXE-CAPS data (Afchine et al., 2018) to detect the clouds. Singer et al. (2021, in prep. same issue) showed good agreement between the cloud detections by both of these instruments.

Remote measurements of cloud backscatter below and above the aircraft were conducted by Miniature Aerosol Lidar (MAL) (Mitev et al., 2002). Backscatter ratios at 532 nm are derived after applying a noise filter, range correction and correction for incomplete overlap in the near range, allowing observations as close as 40 meters from the aircraft.

## 2.2 Satellite observations

The Microwave Limb Sounder (MLS) instrument, operating onboard the NASA Aura satellite, measures various chemical species and temperature and provides over 3500 vertical profiles per day between 82° S to 82° N. Here we use the version 4.2 water vapour profiles described by Livesey et al. (2017), who report for the lower-middle stratosphere a vertical resolution of 3.0 – 3.1 km, horizontal resolution of 190-198 km and an accuracy of 8 – 9%. The data screening criteria specified by Livesey et al. (2017) have been applied to the data. To interpolate the water vapour profiles onto a common potential temperature grid, we use the MLS temperature product provided at the same pressure levels.

Cloud-Aerosol Lidar with Orthogonal Polarization (CALIOP) is a primary instrument onboard the CALIPSO satellite, operational since 2006 (Winker et al., 2009) and providing backscatter coefficients at 532 and 1064 nm with a vertical resolution of 60 m and horizontal resolution of 1000 m in the UTLS. Here we use CALIOP 532 nm level 1B version 4.0 product for diagnosing the cloud vertical cross-sections and for quantifying the cloud top altitude. To enhance the sampling of clouds, we also use level 1 backscatter product at 1064 nm provided by NASA's Cloud-Aerosol Transport System (CATS) lidar operating onboard International Space Station (Yorks et al., 2016). The CATS 1064 nm backscatter is converted to 532 nm using CALIOP color ratio.

The GPS Radio Occultation (RO) technique provides vertical profiles of atmospheric variables with high vertical resolution (~0.5 km around the tropopause), global geographical and full diurnal coverage, and high accuracy (<1 K) (Steiner et al., 1999). We use RO "dry" temperature profiles from COSMIC (Anthes et al., 2008); GRACE (Beyerle et al., 2005) and Metop A/B missions (Luntama et al., 2008) for analyzing the temperature and minimum saturation mixing ratio within AMA during July and August 2017.

## 2.3 Definitions

In this section, we define the key terms regarding the vertical structure o AMA and physical processes therein.

The vertical boundaries of the *tropical tropopause transition layer (TTL)* can be defined using two different approaches reviewed by Pan et al. (2014). The mass-flux approach (Fueglistaler et al., 2009) defines the lower boundary as the tropically-averaged level of all-sky zero net radiative heating (14 km, 355 K) and the upper boundary as 18.5 km

(425 K), where the local mass flux becomes comparable to that of the Brewer-Dobson circulation (Fu et al., 2007). Another approach is based on the TTL thermal structure, where the lower and upper boundaries are defined respectively as the level of minimum stability and the cold point tropopause (CPT) (Gettelman and Forster, 2002). Pan et al. (2014) found that the thermally-defined TTL boundaries are consistent with those derived from the ozone-water vapour relationship. In this study, we adopt the thermal definition of the TTL as in this case the boundaries can be derived from the local instantaneous measurements provided by the Geophysica.,

Since the location of the Geophysica deployment is not tropical in the geographical sense, we refer to the TTL in this region as the *Asian tropopause transition layer (ATTL)* with an upper boundary at the CPT derived from ERA5 temperature profiles collocated with the flight tracks and using airborne temperature profiles. Following Brunamonti et al. (2018), we refer to the upper layer of the Asian anticyclone as confined lower stratosphere (CLS) with a lower boundary at the CPT level and an upper boundary corresponding to the top level of confinement, which they estimate as 63.5 hPa (~440 K) for the 2017 AMA season.

A *convective overshoot* (also termed "ice geyser" by Khaykin et al. (2009)) is defined as detrainment of ice crystals above the local CPT (Danielsen, 1993). Depending on the relative humidity at the level of detrainment, this process can lead either to CLS moistening by rapid ice sublimation, or to irreversible dehydration via uptake of vapour by the injected ice crystals, their growth and sedimentation (e.g. Jensen et al., 2007; Schoeberl et al., 2018). The clouds that have formed in the CLS as a result of local cooling are termed *in situ* cirrus. An *in situ* cirrus cloud is not to be confused with the above anvil cirrus plume (AACP), which is a plume of ice and water vapour in the LS that occurs in the lee of overshooting convection (Homeyer et al., 2017; O'Neill et al., 2021). A *secondary cloud* refers to an *in situ* cirrus that has nucleated from an air mass enhanced in water vapour as a result of convective overshoot.

### 2.4  Ensemble trajectory modeling and convective cloud top data

For investigating the link between the variations in water vapour observed locally by the Geophysica and the deep convection upwind detected using satellite IR imagery, we use the TRACZILLA Lagrangian model (Pisso and Legras, 2008), a modified version of FLEXPART (Stohl et al., 2005). The simulation was designed to release an ensemble of 1000 back trajectories every second along the aircraft flight path, travelling back in time for 30 d. The calculation of back trajectories was performed using the European Centre for Medium-Range Weather Forecasts (ECMWF) ERA5 reanalysis horizontal winds and diabatic heating rates provided at hourly frequency and 31 km horizontal resolution.

For detection of convective cloud encounters (convective hits) along the diffusive back trajectories we use cloud top information from geostationary satellites (MSG1 and HIMAWARI -8). To cover the entire AMA region, we make use of the cloud top product from both the MSG1 images for longitudes west of 90° E and the HIMAWARI-8 images for longitudes east of 90° E. The MSG1 satellite operated by EUMETSAT carries the Spinning Enhanced Visible and Infrared Imager (SEVIRI), providing multi-wavelength image collection with spatial resolution of 1 - 3 km and temporal resolution of 15 min (Schmetz et al., 2002). The Himawari-8 geostationary satellite, launched by the Japan Meteorological Agency (JMA), carries the Advanced Himawari Imager (AHI), providing the images at 0.5 - 2 km spatial resolution with 10 min intervals (Bessho et al., 2016). For computational reasons, we use one image every 20 min.

The cloud top height data was taken from the European Organization for the Exploitation of Meteorological Satellites (EUMETSAT) Satellite Application Facility (SAF) on Support to Nowcasting and Very Short Range Forecasting (NWC) products (Schulz et al., 2009; Derrien et al., 2010, Sèze et al., 2015). The analysis was restricted to the highest and opaque cloud classes that are representative of deep convection. In this study, we consider the convective hits above 100 hPa

 only, corresponding to the cloud tops potentially overshooting the tropopause. They constitute 24.3% of the total number of convective hits identified by this analysis. The convective origin of the sampled parcels is statistically diagnosed in terms of the fraction of convective hits per 1 s sample as well as the convective age of parcel, i.e. the time since convective hit.

It should be noted that the trajectory model integrates 1000 backward trajectories per data point along the flight track which are submitted to a random noise equivalent to a diffusion D=0.1 $m^2\,s^{-1}$ as in Bucci et al. (2020). As such, the integration is a discretization of the adjoint equation of the advective diffusive equation, which is well posed for backward integration (Legras et al., 2005). Unlike single-trajectory Lagrangian calculations, this method does not generate spurious small-scale features as backward time increases, and can be shown to converge with time for a pure passive scalar. With that, the trajectories from each data point come from several, possibly, many sources and the results presented are a statistics over these 1000 trajectories.

Obviously, the results can be affected by biases in the wind field, heating rates and the cloud height product used in this study. The ERA5 is presently considered as the most advanced reanalysis and it was shown to display very consistent transport properties of diabatic versus kinematic trajectories (Legras & Bucci, 2020), which are in excellent agreement with observations (Brunamonti et al., 2018; von Hobe et al., 2021). The main concern in the Asian monsoon region is that ERA5 displays high penetrative convection over the Tibetan Plateau, which might bias the heating rates in the upper TTL over this region (SPARC S-RIP report, 2021). As the trajectories involved in this studied are mostly outside the Plateau, we do not expect any significant impact.

## 3 Evolution of AMA conditions: two modes

### 3.1 Satellite perspective

The 2017 Asian monsoon season was not marked by an anomalous dynamical behavior (Manney et al., 2021), however the campaign occurred during a break - active transition. The strongest convective activity took place in late July and early August above the Southern slopes of Himalayas and the Tibetan plateau  as can be inferred from the low Outgoing Longwave Radiation (OLR) displayed as dashed contours in Fig. 1a. The OLR distribution in July-August 2017 is very similar to its climatological pattern reported by Randel et al. (2015) (cf. their Fig. 5). The thermal conditions across the Asian TTL (ATTL) exhibit a remarkable variability with minimum saturation mixing ratios between 14-18 ppmv in the warmer Northern part of AMA and 2 ppmv above the colder Southern slopes where the Geophysica flights took place. The horizontal distribution of CLS water vapour (390-420 K layer) averaged over 3 weeks before and during the Geophysica flights (Fig. 1b) reveals a pool of moist air with two maxima near the center of the anticyclone.

The time evolution of temperature in terms of $H_2O$ saturation mixing ratio (Fig. 1c) and water vapour (Fig. 1d) within the flight domain shows an interesting development of the UTLS conditions before and during the campaign period. In mid-July, a humid layer in the ATTL starts to build up and propagates above CPT up to about 410 K by early August. The first four Geophysica flights were conducted during this moisture build-up period, characterized by relatively warm CPT temperatures (Fig. 1c), which we term "warm/wet" mode. It should be noted that the 2017 AMA season was marked by a strong positive anomaly in water vapour ranging 1 – 2 ppmv but without a significant tropopause temperature

anomaly (Supplementary Fig. S2). The positive water vapour anomaly is not specific to AMA region and reflects the global wet anomaly in the tropics and subtropics, as revealed by MLS observations (not shown).

In early August, after the warm/wet mode period, the ATTL experienced a rapid cooling and the last four flights sampled a colder and drier ATTL. This "cold/dry" mode is marked by stronger convective activity in the region reflected by low OLR (Fig. 1c), i.e. colder and higher cloud tops, and higher carbon monoxide mixing ratio (Fig. 1d). Since the carbon monoxide is a tracer for troposphere to stratosphere transport, the elevated CO concentration in the LS is indicative of the enhanced upward flux across the tropopause (e.g. Randel et al., 2010.). The cold/dry mode period is also marked by a widespread occurrence of ice clouds above the CPT (377 – 390 K) and as high as 415 K level according to high-resolution cloud profiling by CALIOP and CATS satellite lidars (Fig. 1d). We note that the cold convective period had a transient effect on the CLS water vapour, which mostly recovers the late July values, after the cease of convective activity and tropopause warming in the flight domain by mid-August. A similar inference was reported by Brunamonti et al. (2018) on the basis of balloon soundings in Nepal during the airborne campaign in July-August 2017.

## 3.2   Airborne perspective

The airborne measurements of water vapour and temperature shown in Fig. 2 reflect the satellite-derived development of the UTLS conditions. The ensemble of water vapour profiles obtained using the FLASH hygrometer during the eight StratoClim flights is shown in Fig. 2a. The $H_2O$ vertical profiles at and above the CPT level show a remarkable variability over the two-week campaign period with mixing ratios ranging from 2.8 to 10.2 ppmv. On average, the warm/wet mode yielded an L-shaped mean $H_2O$ profile (solid curve), which is characteristic of the Boreal subtropical conditions, although with notable enhancements at and above the CPT. In contrast, the cold convective period revealed the vertical distribution more typical for the tropical tropopause conditions, with the hygropause at the CPT level. The airborne measurements during both synoptic periods show an accumulation of sharp moist layers above the CPT and up to 410 K level which are diagnosed in the following section. These layers constitute the CLS wet pool seen by MLS, although their sharp vertical structures of sub-kilometer scale cannotbe resolved by the satellite.

The large variability of water vapour is consistent with the tropopause temperature variability, showing minimum saturation mixing ratio between 2.5 and 6.5 ppm and highly variable CPT vertical structure (Fig. 2b), presumably modulated by gravity waves. The CPT potential temperature varied between 370 – 391 K, which is fully consistent with the GPS-RO data.

The highly-variable thermal conditions led to a remarkable dispersion of RHi around the CPT. The clear-sky measurements reveal both a subsaturated and strongly supersaturated environment with RHi spanning 40 – 175 % (Fig. 3a). The credibility of RHi data is ensured by an excellent agreement across the three airborne hygrometers and temperature sensors (TDC and UCSE) (Singer et al., in prep. same issue). In the presence of ice crystals (Fig. 3b), the RHi is generally well above 100% although the subsaturated cloud occurrences were also observed in both dry and wet parts of the water vapour spectrum. Such occurrences are mainly caused by short excursions of temperature above the frost point, which does not necessarily lead to permanent evaporation and depends on the air parcel's (Lagrangian) temperature history. The occurrence of ice crystals was recorded at levels 15 K (~1 km) above the local CPT. The highest-level clouds were detected by the upward-looking MAL lidar at 412 K (18.5 km), which is consistent with NIXE-CAPS detection of cloud particles up to 415 K (Krämer et al., 2020, their Fig. 11) as well as with the maximum cloud altitudes

inferred from satellite lidars (415 K). The presence of ice in supersaturated air is more specific to the cold dry parcels (see also Krämer et al., 2020, their Figure 10d), which suggests a local dehydration during the cold/dry period.

A different perspective on the environmental conditions of cloud occurrence around the CPT is provided in Fig. 3c, showing the distribution of IWC as a function of RHi. The binned ensemble is restricted to the samples, for which both MAS and NIXE-CAPS data indicate the presence of ice particles. The ice crystals found in the subsaturated air above the local CPT are likely to be in the process of sublimation and therefore have a potential for a permanent CLS hydration. Conversely, the crystals in the supersaturated environment will retain their aggregate state and the largest ones (characterized by higher IWC) will sediment down below the tropopause thereby causing permanent dehydration of the CLS. We note that the ice particles in the subsaturated environment account for 14% of the particles detected above the local CPT. This is consistent with a comprehensive analysis of airborne data from various campaigns by Kraemer et al., 2020, who pointed out a significantly larger amounts of IWC in subsaturated ice crystals above the CPT in AMA compared to that in the surrounding tropical regions, which underlines the importance of AMA as the source of LS water

## 4    Convective influence on CLS water vapour

The influence of overshooting convection on the observed water vapour variability was investigated using TRACZILLA ensemble trajectory modeling constrained by satellite cloud imaging and ERA5 reanalysis (see Sect. 2.4). Figure 4a displays a binned ensemble of the measured water vapour mixing ratios color-coded by the convective hits fraction. The trajectory analysis suggests that the convective origin is characteristic to anomalously wet and anomalously dry parcels, which points out the dual role of overshooting convection on the AMA water vapour, i.e. hydration/dehydration. The link to overshooting convection is particularly obvious for a strong water vapour enhancement peaking at 399 K level, which corresponds to F2 of the warm/wet mode (see Sect. 5.1 for detailed analysis of this flight) and for a smaller enhancement at 403 K, corresponding to F7 (see Sect. 5.2). The results for the individual flights are provided in Fig. S3 of the Supplement. The hydrated features (layers of enhanced water vapour, exceeding 1-σ of the campaign ensemble) are denoted throughout the article as *Ax* or *Bx*, where *x* is the flight number.

The elevated convective hits fraction is also characteristic to the driest bins between 370 – 400 K corresponding to F8 of the cold/dry mode with large-scale convection in the flight domain. While the driest parcels are linked with the local or nearby convective events, the wettest ones are traced back to distant convective events all along the circulation pattern of the anticyclone, that occurred several days before their outflows have been sampled by Geophysica. The only exception is F6, which was influenced by a young outflow of a large convective system in the vicinity of the flight (see Sect. 4.2 and Fig. S4 of the Supplement).

### 4.1    Isotopic composition of convective plumes

The relation of moist layers with overshooting convection can be reliably diagnosed using the isotopic ratio of water ($HDO/H_2O$), which is enhanced for water vapour molecules sublimated from ice (Moyer et al., 1996; Hanisco et al., 2007). Figure 4b clearly shows that the wetter parcels in the lower stratosphere are isotopically enhanced, and the wettest of them bear the strongest isotopic signature. This unambiguously points out that the hydrated layers have been produced by overshooting ice geysers. Remarkably, the wet and isotopically enhanced pixels in Fig. 4b are found as high as 420 K level, that is 30 – 50 K above the cold point. Given the diabatic heating rate of 1.1 K/day in AMA and the average

recirculation time of 16 days within the anticyclone (Legras and Bucci, 2020), these hydrated parcels could, in principle, have recirculated twice before being sampled by the aircraft.

### 4.2 Geographical distribution of convective sources

Figure 5 shows the composited map of convective clouds (highest cloud classes), which are linked by trajectories
with the observed hydrated and isotopically-enhanced "wet-and-heavy" parcels and thereby represent the most probable sources of the convectively-processed CLS air sampled by the aircraft. The wet-and-heavy parcels are defined as those located above the local CPT with water mixing ratios exceeding one standard deviation from the median (dashed curve in Fig. 4b) at a given potential temperature level and with isotopic ratios above -400 per mill (except for F1 and F5 where the HDO measurements are not available and the selection is done based on $H_2O$ only). The fraction of wet-and-heavy
parcels to all parcels sampled above the local CPT varies from 0.5% (F6) to 11% (F2) between the different flights. No wet-and-heavy parcels have been detected in F8, which is why it is not displayed in Fig. 5.

The composited map suggests a broad geographical scatter of the convective clouds across the Asian anticyclone. The lifetime of hydrated parcels, as inferred from the back trajectories, ranges from ~12 hours (F6) to about 12.7 days (F7) (see Supplementary Figure S4 for convective age in individual flights). The wet-and-heavy parcels sampled during
the warm/wet mode (F1 through F4) originate from various convective systems in the Northeastern China and Korean peninsula, all of them occurring North of 35 N. The convective age for these parcels varies between 2.6 and 9.9 days. Sensibly, the shortest age corresponds to the lower-height moist layer at 390 K level (F2), whereas the longest age is found for the wet-and-heavy parcels detected as high as at 410 K in F4. The moist features in the flight F2 (A2 and B2, see Sect. 5), found at 390 and 399 K levels, are sourced to different convective events that occurred 2.6 and 4.7 days
before being sampled by Geophysica. We note that while the 1σ-error of the age estimates is generally less than an hour, the attribution of convective sources largely depends on the cloud top data. In particular, the improved v2018.1 trajectory product coupled with NWF SAF geostationary data analysis provided a qualitatively better correlation between the distribution of convective hits and wet-and-heavy parcels as compared with the product used by Bucci et al. 2020.

While the warm/wet mode flights were largely influenced by convection in the Northeastern part of AMA, the wet-
365 and-heavy parcels sampled during flights F5 – F7 are sourced to various different locations. A large convective system over North-Eastern India in the vicinity of the flight on the same day is responsible for the hydration feature in F6 at 380 K level. The convective source of the wet air sampled by F5 is found above Western India, although we note that no isotopic data are available for this flight, whereas the number of parcels with mixing ratio exceeding one standard deviation is small for this flight. In flight F7, the enhanced water vapour features above 400 K (A7 and B7, see Sect. 5)
originate from two different sources: the lower-level feature (A7) is traced back to a group of relatively small systems along the Eastern Chinese coast that occurred 3.9 days before the measurement, whereas the upper one (B7) originates from a large cluster of small-scale convective systems in the center of Asian anticyclone above the northern foothills of Himalayas. We note that this particular region is marked by enhanced water vapour amount according to MLS averages over the campaign period (cf. Fig. 1b). The B7 parcels have thus followed the anticyclonic circulation path for nearly a
full loop before arriving to the flight domain, which took 12.7 days.

The potential for a vapour-rich parcel travelling within AMA CLS to permanently hydrate the stratosphere is determined by the Lagrangian temperature history. We did not analyze the RHi variation along the trajectories, however we quantified the minimum temperatures encountered across AMA using high-resolution GPS-RO profiling. As follows

from Fig. 1a, the subtropical part of AMA has never cooled below the $H_2O$ saturation mixing ratios of around 8 ppmv in July-August 2017, enabling the vapour-rich patches to travel along the northern flank of the anticyclone without freezing. Remarkably, the majority of convective systems identified as the most probable sources of wet-and-heavy parcels (shown in Fig. 5 and marked by black pixels in Fig. 1a) have occurred within the warm tropopause environment in the northern subtropical part of AMA.

It is noteworthy that the probed wet-and-heavy parcels (shown along the flight tracks in Fig. 5) are all located in the northernmost part of the flight domain, i.e. nearer the center of AMA. This is consistent with the spatial distribution of AMA CLS water vapour inferred from MLS (Fig. 1b), showing the maxima above the Tibetan plateau and Sichuan region, that is around the center of the anticyclone. With that, the air circulating near the outer edge of the anticyclone is bound to pass the colder TTL above central India and the Southern slopes, where the organized large-scale convection occurring during the second part of the campaign (cf. Fig. 1) has led to cooling and dehydration around the CPT level. The efficiency of the convectively-induced dehydration, counteracting with the convective moistening in the warmer TTL regions of AMA is considered on a case by case basis in the next section.

## 5    Long-range transport and evolution of moist convective plumes

The hydrated layers in the CLS characteristic of elevated convective hits fraction and/or isotopic enhancement (wet-and-heavy) were detected at altitudes between 16.9 – 19.0 km (380 – 415 K) in all the flights except F8 with the magnitude of water vapour mixing ratio enhancement between 0.9 – 5 ppmv (see Supplementary Fig. S3). The largest enhancement (5 ppmv) was observed in F2 at 399 K (B2 feature), whereas the highest altitude of hydrated layer centered at 18.9 km (411 K) was sampled in F7 (B7 feature). The flights F2 and F7 represent respectively warm/wet and cold/dry modes (see Sect. 3), however in both of these flights the observed moist layers originated from distant convective events. In this section, we provide further insight into the results of F2 and F7 and describe the evolution of the respective moist convective plumes using airborne and satellite measurements.

### 5.1    Warm and wet mode: Flight 2

During the warm/wet mode period, the mean CPT-level water vapour mixing ratio was 7.2 ppmv, whereas the minimum saturation mixing ratio ranged from 5.5 to 6.9 ppmv, according to the airborne data (Fig. 2). During F2, the aircraft was cruising side to side along the Himalayan foothills within Nepali borders gaining altitude in 500 m steps before climbing to 21 km (Fig. 6c). The water vapour vertical profile in Fig. 6a and 6b reveals two layers above the CPT (marked A2 and B2) with water mixing ratio peaking at 10.2 ppmv, twice the campaign-median value at the CPT level. It should be noted that all the three airborne hygrometers report identical spatial structures and absolute values of humidity for these layers, providing full confidence in this observation (see Fig. 3 in Singer et al. (2021), in prep., same issue).

The upper layer (B2) topping at 399 K (~18 km) is characterized by very large fraction of convective hits reaching 0.9 (Fig. 6a) with an average age of 4.7 days (cf. Fig. 5). The convective origin of B2 is unambiguously confirmed by a strong enhancement in the $HDO/H_2O$ ratio of -340 per mill. This is substantially higher than the isotopic ratio found for the equivalent-humidity air below the CPT (about -480 per mill at 373 K level). The enhanced isotopic ratio in this layer clearly indicates that the water vapour enhancement was produced by sublimation of ice. It is remarkable that after nearly 5 days, the convective plume responsible for B2 feature has retained such an amount of moisture.

The underlying wet layer (A2) at ~390 K (~17.5 km) is traced back to a different convective event aging 2.4 days (cf. Fig. 5). However, given that the magnitude of enhancement is nearly the same as that of its upper-level twin, it is conceivable that both A2 and B2 represent the outflow of the same convective event in Northeastern China, and the lower-level A2 feature is a result of gravitational settling of ice crystals shortly after injection.

### 5.1.1 Secondary cloud formation

For an air parcel at 82 hPa bearing 10 ppmv of water vapour (as reported for B2), the saturation is achieved at -78.5 °C. The B2 feature was characterized by the maximum RHi of 116% (Fig. 6c) at -79 °C. At these conditions, a local cooling of 2 °C, which can be produced by a gravity wave (e.g. Kim and Alexander, 2015), would boost the RHi to 165%. This corresponds to the homogeneous freezing threshold at this temperature, hence such a cooling would almost certainly lead to formation of a secondary cloud. Such a cloud was detected by the upward looking MAL lidar at 18 – 18.5 km (398 – 412 K) in F2 with the maximum scattering ratio of 8.1 (marked C2 in Fig. 6c).

Interestingly, the bottom of this cloud is found at the same potential temperature level as the hydrated layer B2 and only about 350 km away from it. Nevertheless, these features appear to have different convective sources. Figure 7a shows the back trajectories released from this cloud intersecting a large convective system above Northeast China (as indicated by red circles with black filling) on 21 July, that is 8 days before F2. The fraction of trajectories intersecting this convective system amounts to 47%. In an attempt to investigate the evolution of humidity of this air mass, we searched for the MLS swaths collocated in space and time (within 500 km and 1 hour) with the tracked parcels. A perfect match was found on 24 July: the MLS swath lies precisely across the cluster of the tracked parcels as shown in Fig. 7b. The nearest MLS profile reports 8 ppmv at the parcel level, which is 2 – 3 ppmv wetter than the neighboring measurements along the same orbit. This suggests that the moist plume remained compact (at least in the meridional plane) up to 3 days after the convective event.

The Lagrangian temperature history of this air mass (Fig. 7c) suggests that since the convective encounter, the parcels remained subsaturated most of the time and, in particular, during the collocated measurement by MLS. The RHi was estimated from the ERA5 temperature and pressure along the back trajectories, whereas the mixing ratio was assumed to be constant 12 ppmv. The episodes of moderate supersaturation with RHi reaching 140% were encountered between about 144 to 170 h before the sampling and it is conceivable that cirrus could have formed around that time and some water was lost to sedimentation. However, the episodes of strong supersaturation with RHi reaching the homogeneous freezing threshold were encountered only during the last day before the measurement, when the parcels were entering the colder CLS above the Southern slopes. The RHi along the back trajectories during the last day was reaching 160%, which would enable ice nucleation and repartitioning of the excessive vapour into a secondary cloud.

### 5.2 Cold and dry mode: Flight 7

The cold/dry mode was marked by a synoptic-scale cooling throughout the 370 – 400 K layer extending across the CPT. The largest vertical extent of the cold layer was observed in F7, where the saturation mixing ratio dropped below 4 ppmv throughout 370 – 397 K layer (cf. Fig. 2b). The northbound flight leg of F7 (see flight track in Supplementary Fig. S1) included several porpoises across the CPT level (varying between 375 – 383 K) as shown in Fig. 8a. The time series of potential temperature is marked with ice particle occurrence detected by MAS, which shows the presence of subvisible cirrus clouds (color tagging) with scattering ratio below 6 extending up to 400 K level.

The water vapour time series in Fig. 8a reveals a remarkably large horizontal variation of mixing ratio in this layer, spanning 3.0 to 6.2 ppmv on a horizontal scale of one hundred kilometers. Almost the entire CPT-porpoising segment of F7 shown in Fig. 8a (20000 – 22200 s) is supersaturated with RHi reaching 155%, whereas the water vapour variation follows the saturation mixing ratio with a high degree of correlation (r = 0.97). The occurrence of ice particles detected by MAS is reflected by enhancements in IWC shown as blue shading in Fig. 8a,b. The magnitude of IWC enhancements (up to 3.3 ppmv) is comparable to the magnitude of water vapour reduction, which suggests that these ice crystals have formed in situ as a result of synoptic-scale CPT cooling. Indeed, as shown in Fig. 8b, the total water does not exceed the background level represented by the mean water vapour from the previous flights. It is however still possible that some of these crystals were produced by overshooting as suggested by Lee et al. (2019) for this particular flight.

Above the layer of thin cirrus reaching 400 K level, the water vapour profiles in F7 reveal two enhancements, marked in Fig. 8 as A7 and B7. The B7 feature is characterized by a maximum enhancement of 1.9 ppmv at 410 K in a layer extending between 405 – 415 K (18.5 – 19 km). Both A7 and B7 moist features are characteristic of significant isotopic enhancement (Fig. 8c), whereas the B7 is also marked by an enhanced fraction of convective hits (cf. Fig 4a and Supplementary Fig. S3). From the convective sources' analysis in Sect. 4.2, we know that the hydrated feature B7 has a convective age of 12.7 days during which the moist convective plume has made a nearly complete circle within AMA. During this time, the mixing within the moist layer is expected to smoothen its vertical structure, however the B7 enhancement reveals a rather sharp vertical structure. Such a sharp structure is normally associated with recently sublimated ice crystals from a nearby overshoot (e.g. Khaykin et al., 2009; 2016). The absence of recent (< 5 days) convective events (see Supplementary animation) upwind of B7 has led us to investigate the satellite cloud measurements and temperature history along the corresponding backward trajectories.

### 5.2.1 Secondary cloud sublimation

Figure 9a shows an ensemble of back trajectories released from B7 together with the ground tracks of CALIPSO and CATS nighttime orbits nearest in time and space with the location of the sampled air parcels along their trajectories. The locations and timing of satellite lidar transects were favorably close to the locations of the tracked parcels at each given time: the largest temporal offset between the trajectory time and a lidar transect is only about 3 hours. This warrants investigation of the Lagrangian evolution of cloud occurrence in the B7 parcel.

On 6 August, neither CATS nor CALIOP transects (marked as T1 and T2 in Fig. 9a) show the presence of clouds above 18 km, which is consistent with the parcel's temperature history in Fig. 9c showing sub-saturated conditions before T1. After passing the T2 point, the parcel has experienced a strong cooling episode, boosting the maximum RHi above the homogeneous freezing threshold. The next collocated lidar overpass (T3) took place on 7 August when the parcel's temperatures have just relaxed down to saturation levels. The lidar curtains labeled T3 and T4 (Fig. 9b) show evidence of partial evaporation of the cirrus cloud cross-sampled by CATS at 17 UT (T3) and by CALIOP three hours later (T4). The location of the tracked parcels (marked by a white rectangle in Fig. 9b and Supplementary Fig. S5b), matches precisely the evaporating fraction of the cloud. Thus, the final sublimation of this secondary cloud has occurred about 13 hours before B7 sampling, which can explain its sharp vertical structure.

At the time of B7 sampling, the parcel's RHi – computed from ERA5 temperatures and FLASH peak value of 7.3 ppmv in the hydrated layer – amounts to nearly 100%, which is consistent with the airborne temperature measurement. Downwind of the flight track there are two transects (T5 and T6), not necessarily collocated in time with the westward

progression of B7 parcel, but showing an absence of ice particles at the respective level (Fig. 9 and Supplementary Fig. S5).

The above led us to conclude that while the B7 water vapour enhancement was produced by a 12.7-days old convective plume that circumnavigated AMA, its vertical structure was modified by strong yet transient cooling episodes that acted to temporally repartition the vapour into ice on a scale of several hours. As inferred from NIXE-CAPS particle
size distribution measurements in F8, a freshly nucleated in situ cirrus near the CPT level is dominated by very small ice crystals with effective diameter of 4 -10 μm (Supplementary Fig. S6). According to Muller and Peter (1992) such crystals would sediment at a rate of 0.6 - 2 cm/s. Assuming the onset of ice crystals nucleation at B7 - 20 h (corresponding to the onset of the strong cooling episode) and their evaporation at T4 point, the cloud particles should have sedimented by less than 200 - 700 meters during their lifetime.

With this case we point out that the homogeneously-nucleated crystals smaller than 10 μm occurring in the CLS as a result of convectively-induced radiative cooling and/or gravity waves-induced temperature perturbations do not last long enough to sediment out from the stratosphere and therefore have limited potential to dehydrate the CLS.

## 6    Discussion and summary

The occurrence of water vapour enhancements in the lower stratosphere associated with overshooting convection
has been reported in several studies based on in situ measurements in the deep tropics over Western Africa (Khaykin et al., 2009; Schiller et al., 2009), Northern Australia (Kley et al., 1993; Corti et al., 2008), South America (Khaykin et al., 2013), Central America (Sargent et al., 2014), Western Pacific (Jensen et al., 2020) as well as at midlatitudes over North American monsoon (Hanisco et al., 2007; Weinstock et al., 2007; Smith et al., 2017) and Asian monsoon (Vernier et al., 2018; Brunamonti et al., 2018; Krämer et al., 2020).  We note that the reported cases represent a small fraction of in situ
measurements acquired; there is typically no more than one case of water vapour enhancement above the tropopause detected during a given field campaign.

Compared to other field campaigns, the StratoClim aircraft deployment in Nepal provided an ample sampling of moist layers above the tropopause. Their convective overshooting origin is unambiguously supported by both the enhanced isotopic ratios in the moist plumes and by their traceability to convective events. Notably, the occurrence of
lower stratospheric moist plumes above the monsoon regions is also supported by satellite observations (Fu et al., 2006; Schwartz et al., 2013; Werner et al., 2020), whereas the enhanced water isotopic ratios observed over these regions (Hanisco et al., 2007; Randel et al., 2012) support the role of overshooting convection in maintaining the water vapour maximum in the North American and Asian monsoon anticyclones. This process adds to the radiatively-driven slow ascent of wet air through the warm tropopause in the northern part of AMA. Another possible pathway of water into the CLS in
addition to the slow ascent and overshooting may be the isentropic transport across the CPT from the Tibetan plateau (characterized by highest CPT) to the southern slopes of Himalayas.

Using MLS observations and OLR data, Randel et al. (2015) concluded that stronger convection in the Asian monsoon region leads to colder and drier lower stratosphere whereas the opposite is true for the weaker convection. They also pointed out the importance of subseasonal variations of deep convection driving the water vapour amount near the
tropopause. Interestingly, the composited maps of OLR anomalies for wet and dry modes (Fig. 5 in Randel et al. (2015)) reveal an east-west dipole and in both cases this dipole is centered exactly on the StratoClim flight domain. Furthermore,

the evolution of the UTLS conditions in the flight domain, switching from warm/wet to cold/dry mode over the course of the campaign, allowed for sampling the opposite-sign effects of deep convection on the water vapour above the tropopause.

The warm/wet mode sampled during the early flights revealed substantial enhancements of water vapour mixing ratio reaching above 10 ppmv (twice the background) as high as 400 K (18.2 km) level, but very little evidence for dehydration upstream. By contrast, the second (cold/dry mode) period of the campaign with organized large-scale convection inside and close to the flight domain led to synoptic-scale CPT cooling and a drastic drop of water vapour by ~30% near the tropopause. We note though that the dehydration layer did not extend above 395 K, whereas in the upper layers, the excess of water vapour was subject to a transient phase transition, resulting in an outbreak of cirrus at levels up to 415 K (18.9 km)  A similar inference was made by Brunamonti et al. (2018) on the basis of balloon soundings of water vapour and ozone in Nepal as part of StratoClim campaign in 2017. They argued that overshooting convection is responsible for an isolated maximum of $H_2O$ in the CLS observed in July 2017, whereas the water vapour minimum at the CPT level is caused by synoptic-scale cold anomaly above the southern slopes that maximized around 9 August.

Our trajectory analysis suggests that convective origin is characteristic of the wettest and the driest parcels (Fig. 4a), which points out the dual role of overshooting convection on the AMA water vapour. With that, we note that the probability of dehydration decreases with the age of convective outflow, ascending within AMA at an average 1.1 K/day rate in potential temperature (Legras and Bucci, 2020). This way, a hydrated air mass, circulating within the confined anticyclone, progressively moves up and away from the tropopause and becomes less likely to encounter permanent dehydration. A similar inference was made by Ueyama et al., (2018) on the basis of trajectory-based microphysical simulations.  Although, as we showed here, the secondary clouds can form as high as 410-415 K level, their lifetime is limited to a fraction of day, which does not enable a permanent removal of water vapour from the CLS. Thus, a hydrated plume that survived a full turnover within AMA would retain its moisture and eventually loft it into the free stratosphere.

The question on the role of different AMA sub-regions in the cross-tropopause transport of water has been addressed by a number of studies quoted in the introduction. With that, there appears to be no consensus regarding the dominance of a particular sub-region. In this study, the majority of the observed wet plumes in the CLS are traced back to convective events in the northeastern part of AMA, which influenced the flights during the first (warm/wet) period of the campaign. The other flights have sampled wet air originating from convection above the Tibetan plateau as well as northeastern and northwestern India. Given the limited time period of the campaign and the large subseasonal variability of the Asian monsoon, this inference may not be fully representative of the climatological convective source regions. Nevertheless, it can be concluded that convection occurring in the northern and northeastern parts of AMA, characterized by a warmer tropopause, is more likely to produce persistent moistening of the lower stratosphere.

The airborne measurements in AMA within StratoClim have revealed the abundance of moist convective plumes in the CLS. In this respect, the Asian anticyclone is very similar to its North American counterpart. Indeed, both anticyclones extend well into the extratropics, where a warmer tropopause enables unimpeded transport of large amounts of water vapour. An important finding of our study is the persistence and recirculation of moist convective plumes in the confined LS of AMA. To our knowledge, such phenomena were never before observed in the deep tropics. The recirculation of water vapour-enhanced air masses was reported in the Antarctic and Arctic vortices (Voemel et al., 1995; Khaykin et al., 2013) where the hydration of lower stratosphere occurs through sedimentation of ice PSCs.

Overall, our results suggest a complexity of processes controlling water abundance and its aggregate state in the lower stratosphere of AMA. The strong isotopic enhancements specific to the moist layers in the CLS and their traceability to convective events consistently suggest that overshooting convection is an important contributor to the seasonal maximum of water vapour in the AMA lower stratosphere. At the same time, the large-scale organized convection in the southern part of AMA is shown to cause synoptic-scale dehydration around the tropopause through radiative cooling. Another mechanism of dehydration is the overshooting of ice crystals into the supersaturated environment above the tropopause, which leads to their rapid growth and sedimentation. The evidence of such a process was obtained in a particular flight (F8) and will be a subject of a separate study.

Further insights into the AMA gaseous/particular composition and dynamics will be provided by an upcoming airborne campaign within the Asian summer monsoon Chemical and Climate Impact Project (ACCLIP; https://www2.acom.ucar.edu/acclip), which will sample the Western Pacific mode of the monsoon and eastward eddy shedding using NASA WB-57 and NCAR GV aircrafts. The stratospheric impact of overshooting convection in the North American monsoon is a primary target of the Dynamics and Chemistry of the Summer Stratosphere (https://dcotss.org/) project, involving ER-2 high -altitude aircraft.

*Acknowledgements*

We gratefully thank the StratoClim coordination team, and the Myasishchev Design Bureau for successfully conducting the field campaign. This work was supported by the StratoClim project by the European Community's Seventh Framework Programme (FP7/2007–2013) under grant agreement no. 603557 and the TTL-Xing ANR-17-CE01-0015 projects. Meteorological analysis data are provided by the European Centre for Medium-Range Weather Forecasts. ERA-5 trajectory computations are generated using Copernicus Climate Change Service Information. We also thank the AERIS/ICARE data and service centre for providing access to the MSG1 and Himawari data and computer resources for the production of the cloud top product using SAFNWC GEO-v2018.1 algorithm. Last but certainly not least, we sincerely thank the three anonymous referees for their constructive remarks.

**Author contributions**

SK performed the airborne and satellite data analysis and wrote the draft. EM and BC provided airborne water isotopes data. SB and BL performed trajectory calculation and geostationary satellite data analysis. AL, IF and VY provided airborne water vapour data. MK, AA, CR and NS provided airborne total water and ice particles size data. FC provided airborne particle backscatter in situ data. VM and RM provided airborne lidar data. VV provided airborne temperature data. EM, CS, BC, MK, CR, BL and FS provided useful comments and participated in the redaction of the paper.

**Data availability**

The airborne data will be available from the HALO database at https://halo-db.pa.op.dlr.de/mission/101 (last access: 30 July 2021) (German Aerospace Center, 2021), in the meantime they may be provided by respective PI upon request. TRACZILLA data are available upon request. MLS data are publicly available at http://disc.sci.gsfc.nasa.gov/Aura/data-holdings/MLS; GNSS-RO data at https://www.romsaf.org/product_archive.php; CALIOP data

at https://doi.org/10.5067/CALIOP/CALIPSO/CAL_LID_L1-VALSTAGE1-V3-40; CATS data at
https://cats.gsfc.nasa.gov/data/

**Competing interests**

The authors declare that they have no conflict of interest.

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

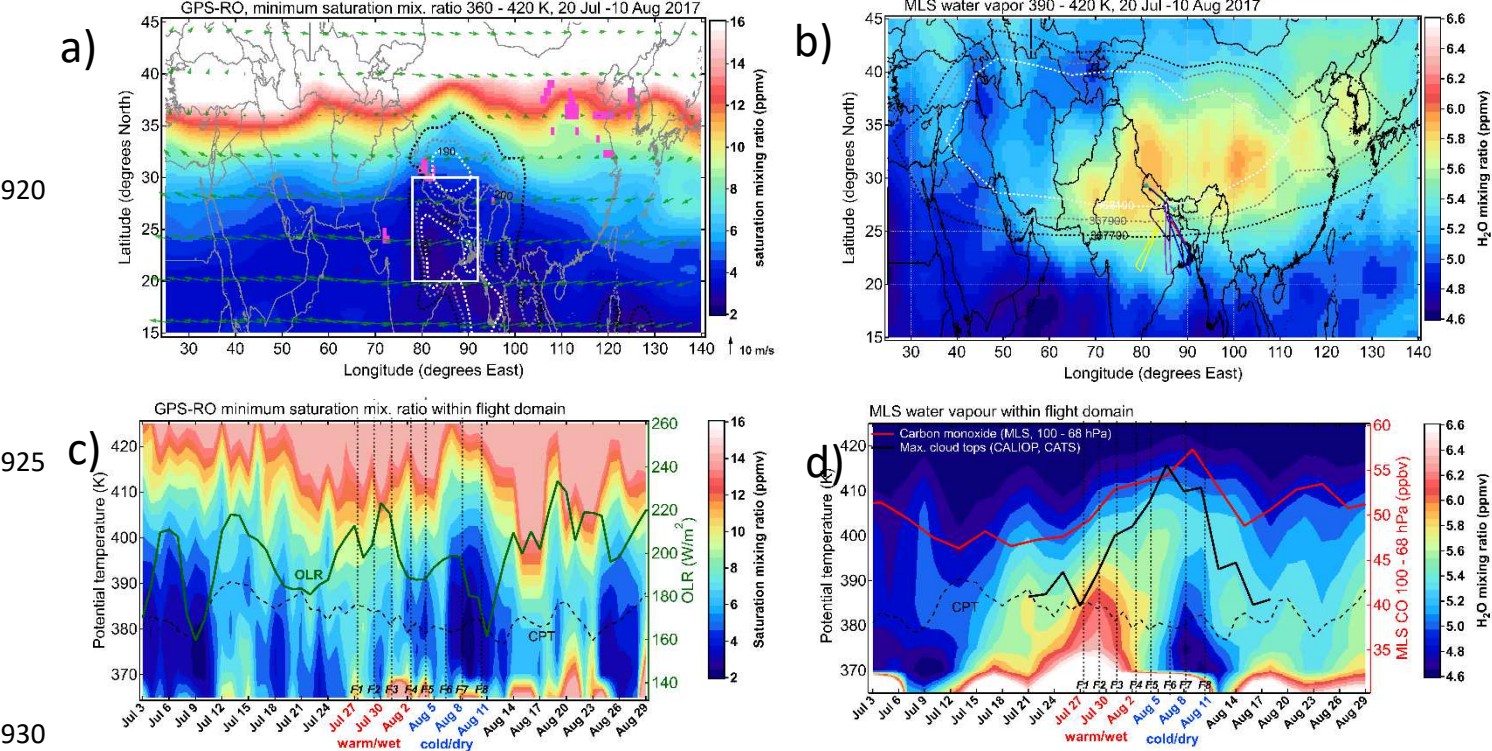

**Figure 1. Overview and evolution of AMA UTLS conditions during July-August 2017 from satellite observations. (a) Color map shows GPS-RO minimum saturation mixing ratio (ppmv) encountered within 360 – 420 K layer during 20 July – 10 August period. Dashed contours indicate NCEP OLR (W/m2) in the range 190 – 210 W/m2 (contour interval 10 W/m2). Green arrows are wind velocity vectors (ERA5, 70 – 100 hPa average for the campaign period). Pink pixels mark the most probable sources of the hydrated features (see. Sect. 4.2). White rectangle indicates the flight domain. (b) Horizontal distribution of MLS water vapour in the confined LS (390-420 K layer) averaged over 20 July – 10 August period with the date-colored flight tracks superimposed (see Fig. 2 for color definition). Dashed contours depict Montgomery stream function (m$^{-2}$s$^{-2}$), which is used to define the horizontal boundaries of AMA at 390 K – 410 K levels (Santee et al., 2016). (c) Evolution of GPS-RO minimum saturation mixing ratio profile within the flight domain (78 °E – 92 °E, 20 °N – 30 °N). Green curve plotted versus right-hand axis depicts the domain-mean OLR. (d) Evolution of MLS water vapour profile within the flight domain. The solid black curve depicts the maximum level of cloud tops detected using CALIOP and CATS satellite lidars. The red curve indicates MLS carbon monoxide (68 – 100 hPa mean). The vertical dashed lines indicate the flight dates (Flight number is given as *Fx*) and their vertical extent, the dashed curve marks the average CPT potential temperature level (GPS-RO) in both (c) and (d).**

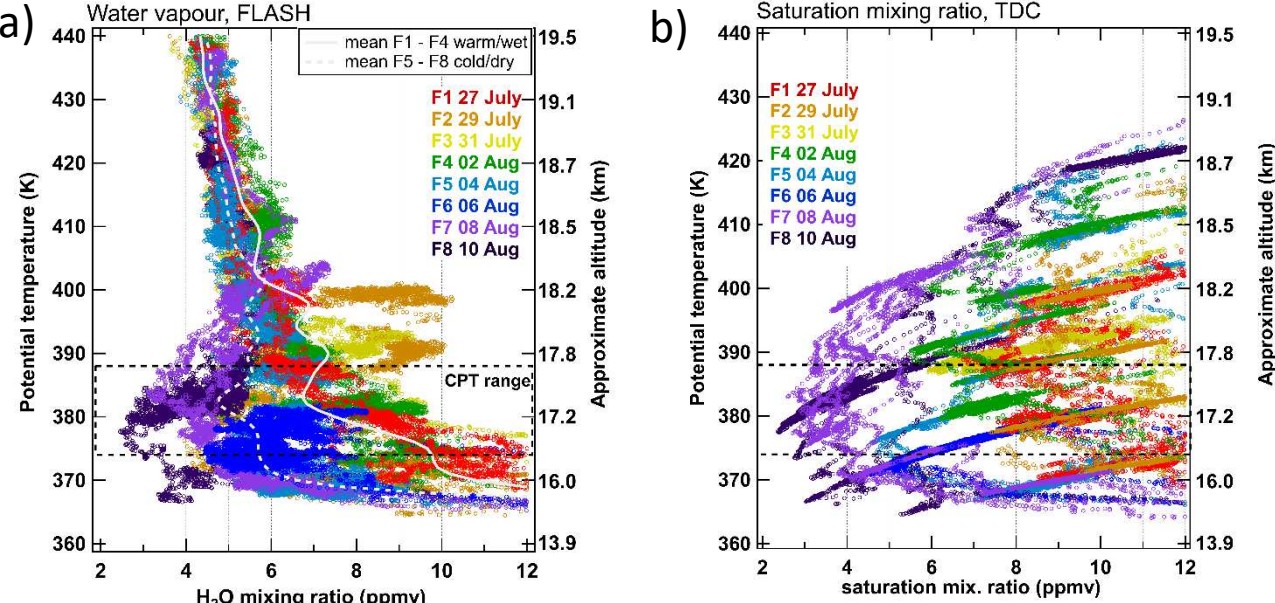

**Figure 2. (a)** Water vapour mixing ratio profiles taken by FLASH hygrometer. The flight dates are shown in the legend. The dashed rectangle marks the CPT vertical range. The white solid and dashed lines depict the mean water vapour profile for the warm/wet and cold/dry modes respectively (see Fig. 1c, d). **(b)** Saturation mixing ratio profiles computed form TDC measurements of temperature and pressure. The horizontal dashed lines in both panels mark the vertical range of CPT level encountered during the campaign. Note the strong variability of water vapour at the CPT level and accumulation of sharp enhancements in the lower stratosphere.

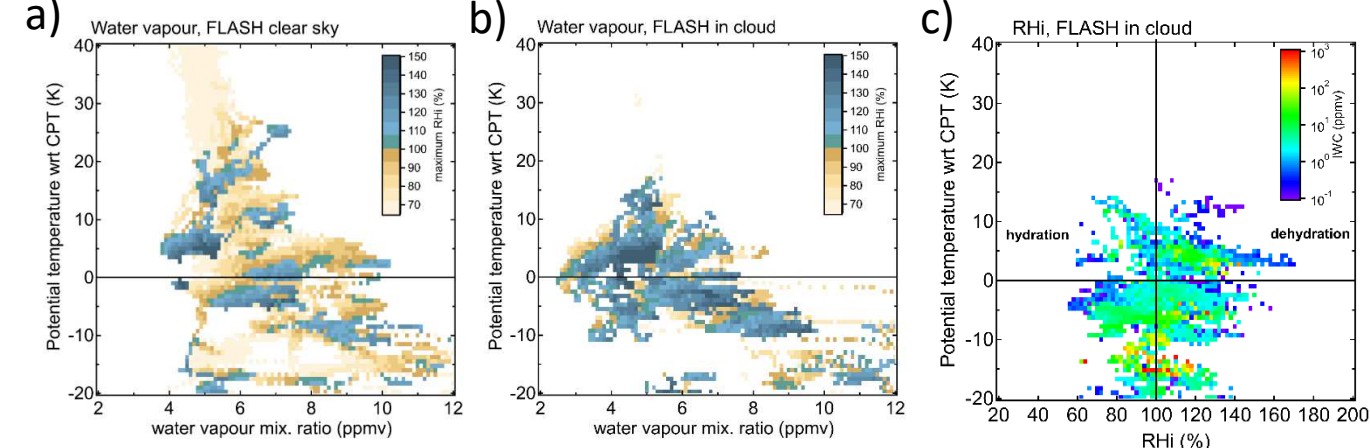

**Figure 3. (a) and (b)** Binned distribution of maximum RHi (computed from FLASH and TDC measurements) as a function of FLASH water vapour and potential temperature relative to the local CPT level (bin size 0.1 ppmv by 1 K) for **(a)** clear sky conditions and **(b)** cloud occurrence detected by MAS (see Sect. 2.1.2). **(c)** Binned distribution of ice water content (IWC) exceeding 0.1 ppmv from NIXE-CAPS instrument inside clouds detected by MAS as a function of relative humidity over ice (RHi) and potential temperature relative to the local CPT level. The presence of ice crystals in subsaturated air (RHi<100%) above the cold point tropopause potentially leads to permanent hydration of the CLS, whereas in supersaturated air the ice crystals are expected to sediment out of the CLS, thereby leading to its dehydration.

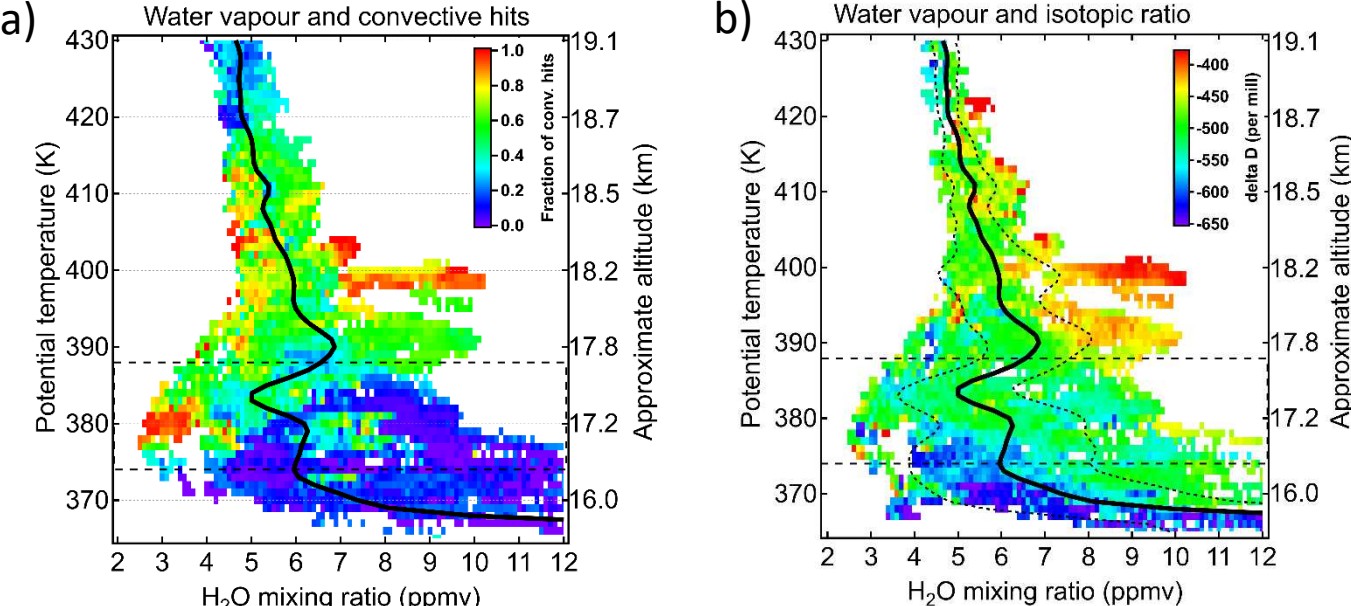

**Figure 4. Binned ensemble (bin size 0.1 ppmv by 1 K) of FLASH H₂O measurements from all flights except F1 and F5 for which no isotopic data are available. The pixels are color-coded by (a) convective hits fraction (see text for details) and (b) ChiWIS HDO/H2O isotopic ratio. The black solid and dashed curves depict campaign-median H2O profile and one standard deviation respectively (all flights). The horizontal dashed lines mark the vertical range of CPT level encountered during the campaign. Note that the convective origin is specific to both anomalously wet and anomalously dry parcels (a), whereas the wettest parcels in the lower stratosphere are isotopically enhanced (b).**

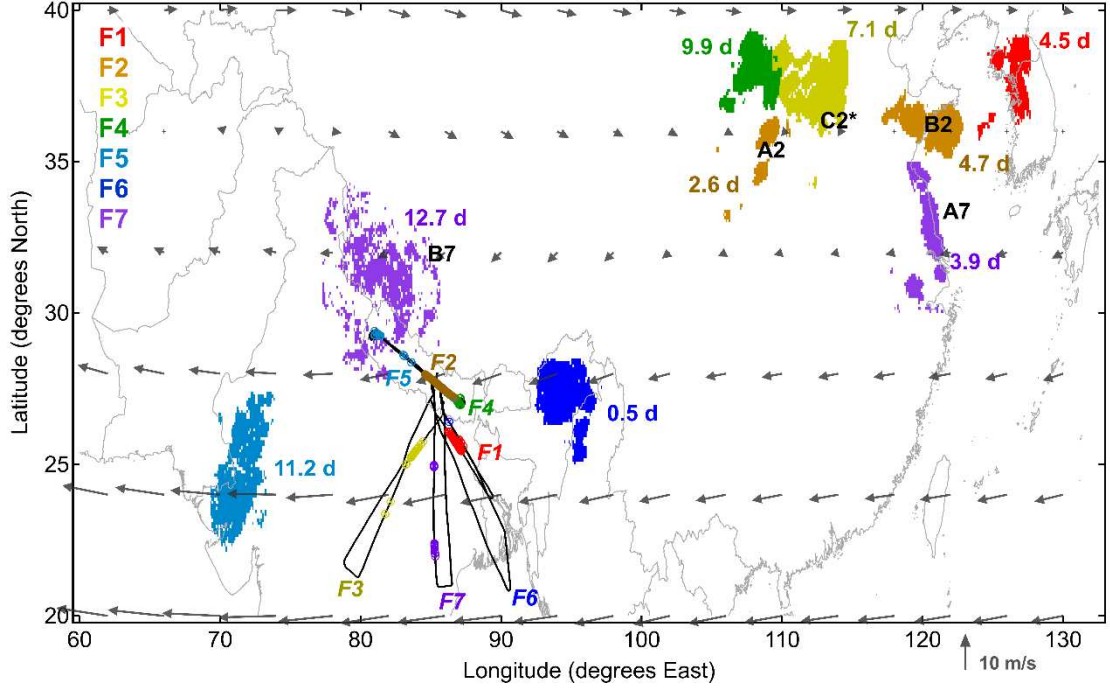

**Figure 5. Composited map of convective clouds (highest cloud class) linked by back trajectories with the observed hydrated (wet-and-heavy) parcels. The color of convective systems indicates the flight number (*Fx*, see legend) in which the respective parcels were sampled. The same color code is used to mark the flight segments where these parcels were probed. The convective age (days) is indicated for each convective system. The convective sources for the features of interest in flight F2 (A2, B2, C2*) and F7 (A7, B7) are annotated (see Sect. 5). The arrows are wind velocity vectors (ERA5, 70 – 100 hPa average).**

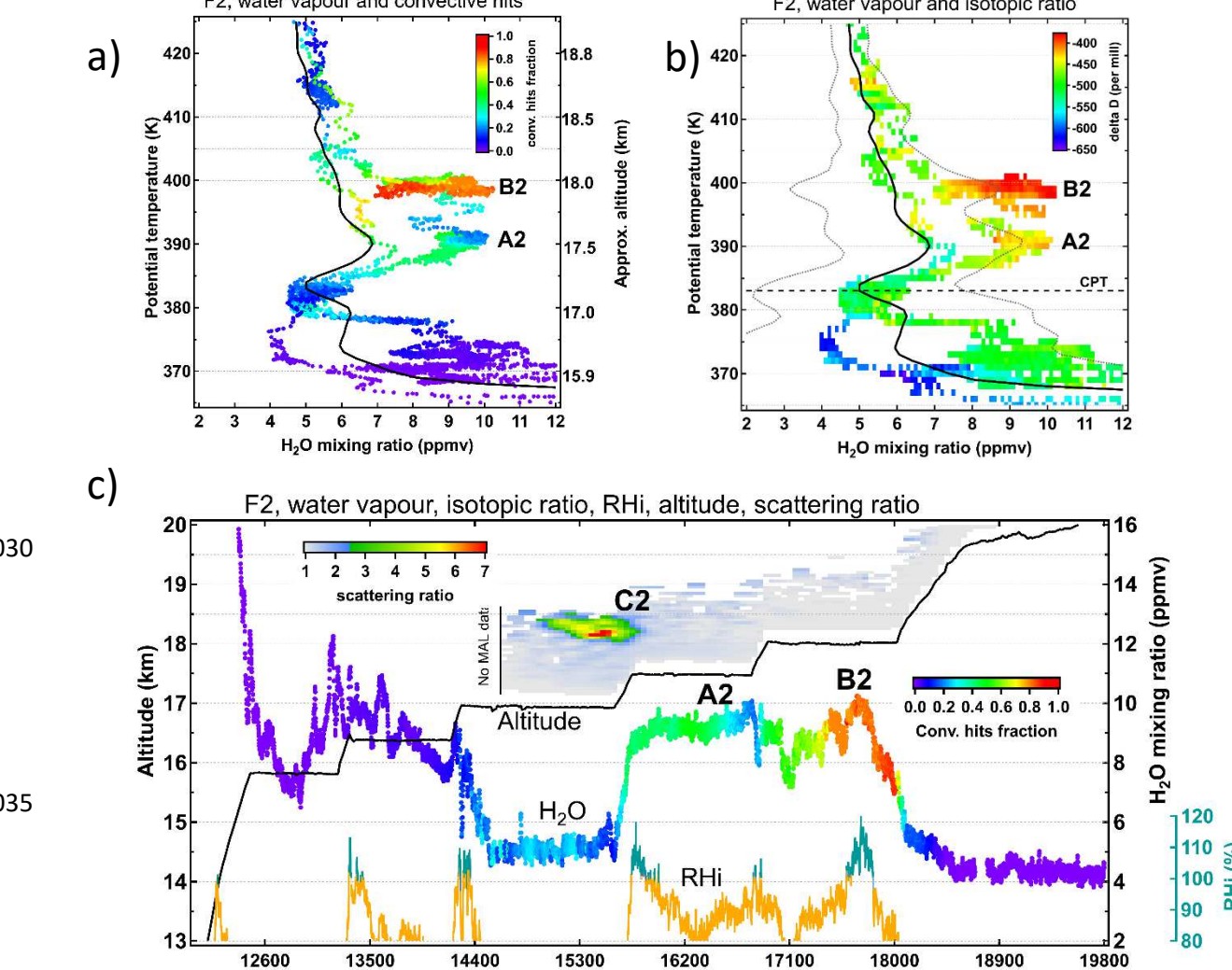

**Figure 6. Results of F2 measurements (29 July) with the features of interest marked A2, B2 and C2. (a) water vapour profile (FLASH) color-coded by convective hits density (see text for details). (b) same as (a) but color-coded by ChiWIS HDO/H2O isotopic ratio. The black solid and dashed curves depict campaign-median H2O profile and two standard deviation respectively. (c) Time series of flight altitude (black curve), water vapour (color-coded by convective hits density) and RHi (right-hand axis) computed from FLASH and TDC measurements. The color map shows scattering ratio measured by MAL (upward-looking).**

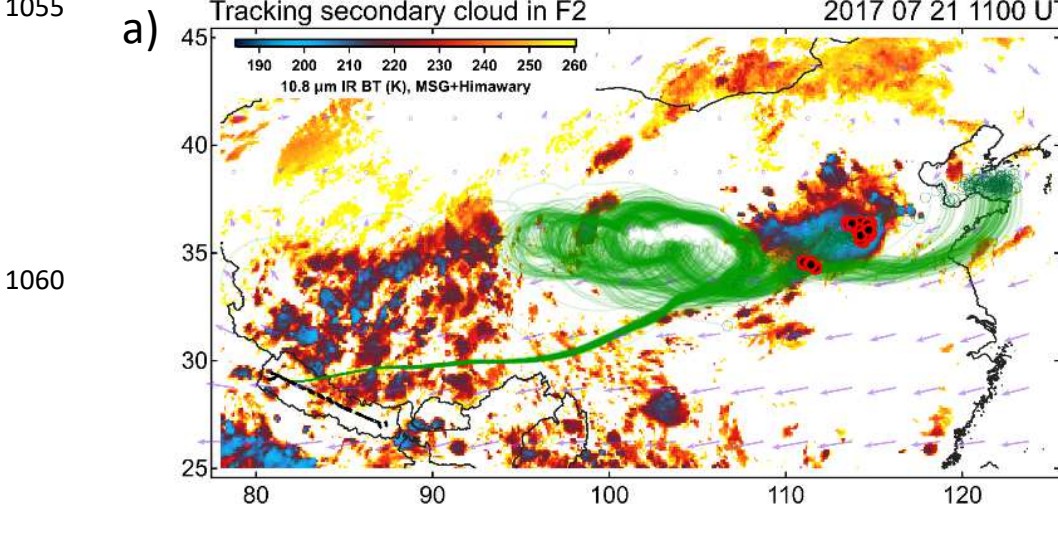

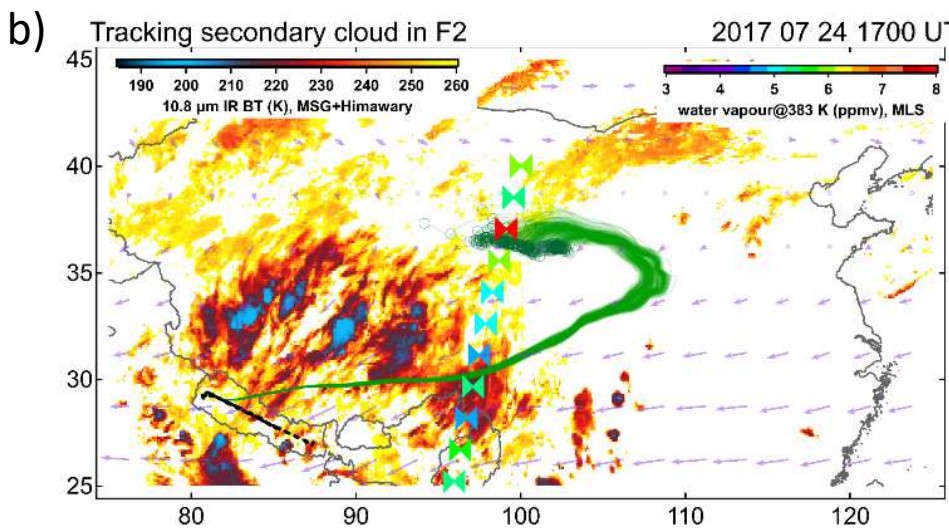

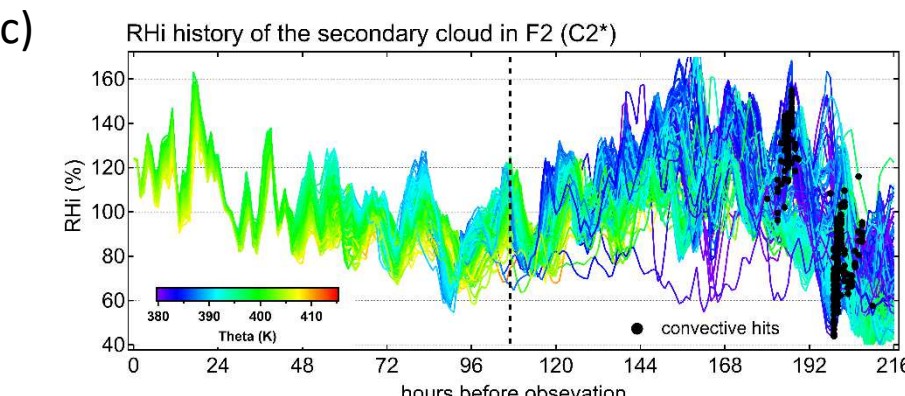

**Figure 7. Trajectory analysis of the secondary cloud (C2) detected in F2 (29 July). (a) Back trajectories (green curves) and**
1085 **convective hits locations (black-filled red circles) superimposed onto the IR BT at the time of convective hits (cf. time stamp in the panel). (b) Same as (a) but for the time of the MLS sampling of the moist plume. The spatiotemporally-collocated MLS swath is displayed as markers color-coded by H2O mixing ratio at 393 level. The wettest MLS measurement coincides with the location of hydrated parcels. (c) Relative humidity over ice evolution along the back trajectories with color coding by potential temperature computed from ERA5 temperature and assumed H$_2$O mixing ratio of 12 ppmv (see text for details). The balck**
**markers show the locations of convective hits, the vertical dashed line indicate the time of MLS sampling of the moist plume on 24 July.**

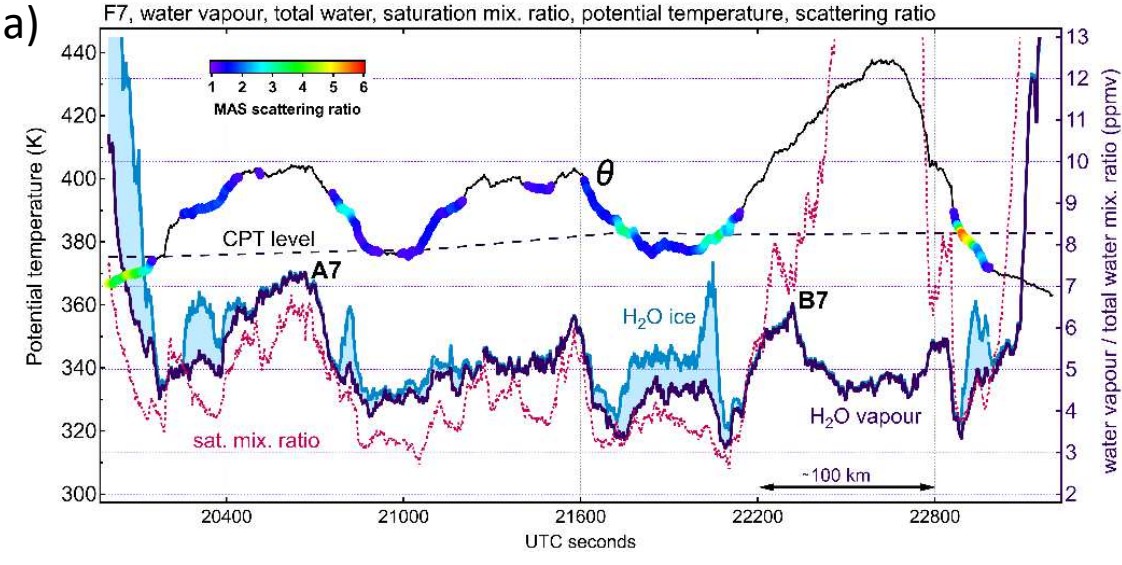

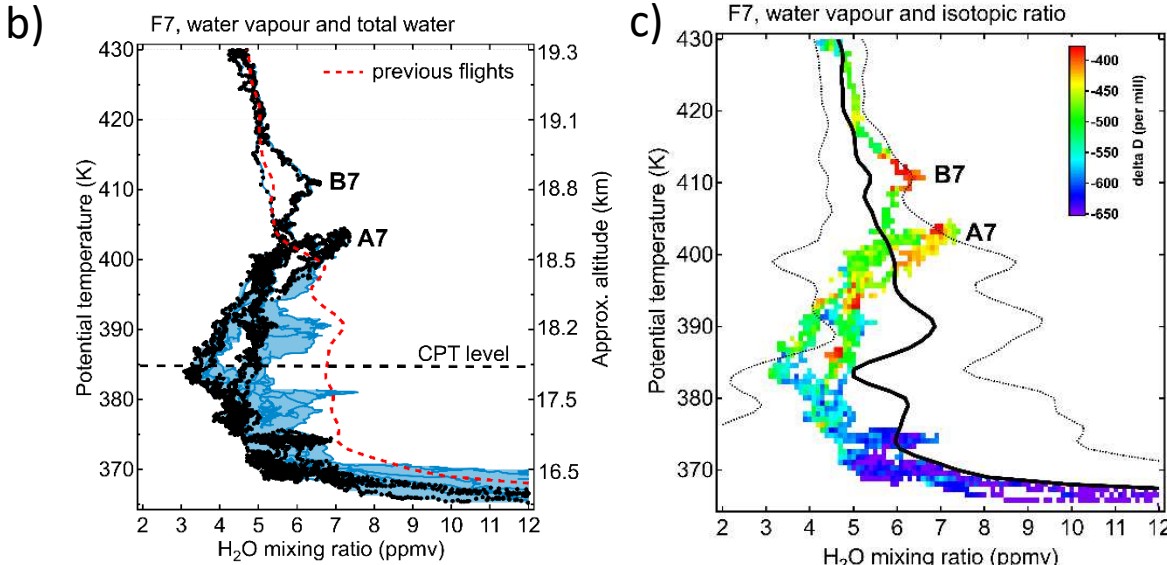

**Figure 8. Results of F7 measurements (08 August) with the features of interest marked A7 and B7. (a) Time series of potential temperature $\theta$ (left-hand axis) and water vapour/total water (right-hand axis). The $\theta$ time series is tagged by cloud occurrence and color-coded by scattering ratio. The IWC is shown as cyan shading stacked on the water vapour curve, the darker cyan curve depicts the total water (right-hand axis). The dashed magenta line depicts the saturation mixing ratio (right-hand axis).**

**The CPT level displayed as black dashed line is inferred from airborne measurements. (b) Vertical profiles of water vapour (black circles), total water (cyan) and IWC (cyan shading). The red dashed curve indicates the mean water vapour profile form the previous flights (F1 - F6). (c) Binned water vapour profile with pixels color-coded by HDO/H2O isotopic ratio. The black solid curve depicts the campaign-median H2O profile, the dotted curves represent two standard deviations respectively.**

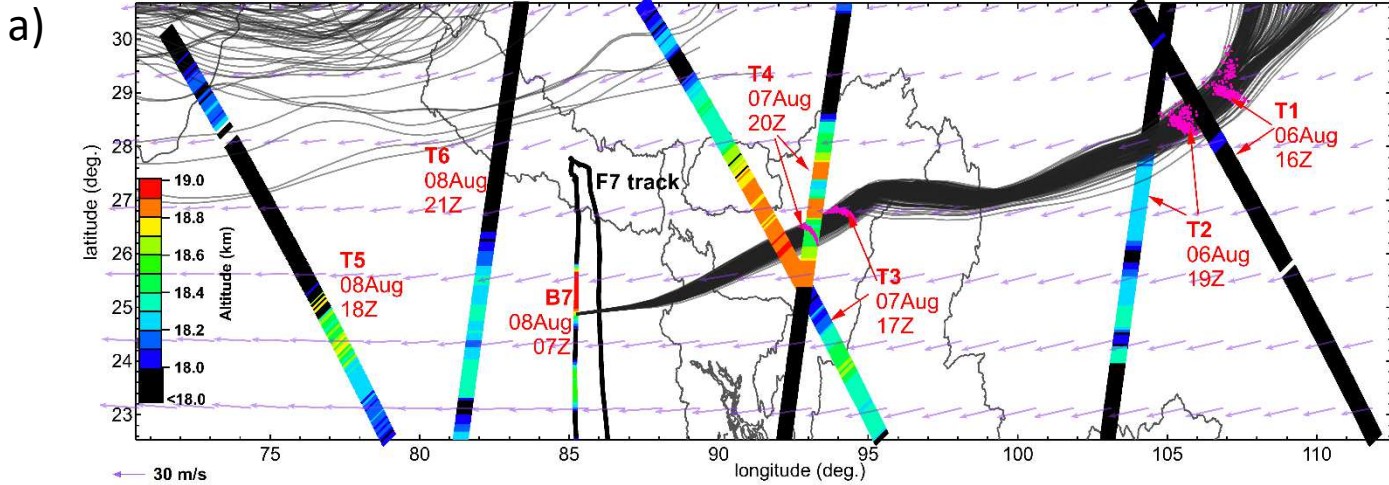

a)

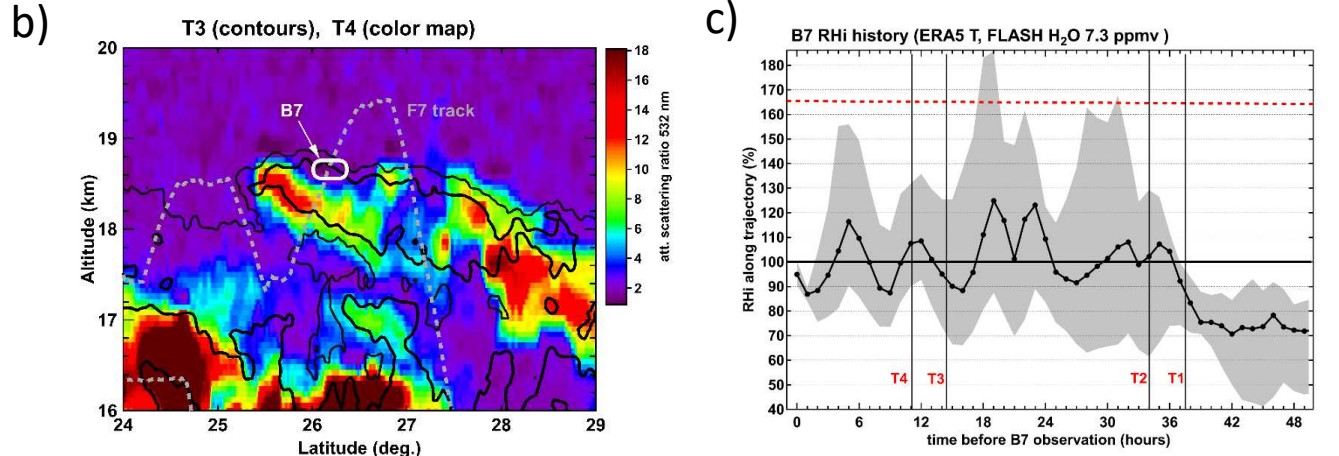

b)

c)

Figure 9. Evolution of cloud occurrence and RHi along the backtrajectories released from B7 (H2O enhancement at 410 K in F7). (a) Backtrajectories (black curves) and ground tracks of CALIPSO and CATS satellite lidars color coded by the cloud top altitude. The magenta circles mark the locations of the tracked parcels at the time of the nearest satellite transacts. The red labels indicate the transacts number (T1 – T6) and its UTC. The arrows are wind vectors (ERA5) interpolated at 410 K level.
The track of F7 flight is shown as solid curve with altitude color-coding. (b) Attenuated scattering ratio (SR) for T3 (contours, 3 and 6 SR units) and T4 (color map). The white oval shows the latitude-altitude location of the B7 parcels at the time of T4 transect. The grey dashed curve is the projection of F7 flight track onto the lidar section. Note that the location of B7 parcels matches the evaporating part of the cloud. (c) Evolution of RHi along the backtrajectories computed using ERA5 temperatures and measured H2O mixing ratio in B7 (7.3 ppmv). The shading indicates two standard deviations, the vertical lines with red
labels mark the timing of satellite transacts, the red dashed line shows homogeneous freezing threshold for the given temperature and humidity.