# Peer review of "Persistence of moist plumes from overshooting convection in the Asian monsoon anticyclone"

_Atmospheric Chemistry and Physics, 2021_

## Referee Comment (RC2)

Review of **Persistence of moist plumes from overshooting convection in the Asian monsoon anticyclone** by Sergey Khaykin et al.

The authors present a thorough and comprehensive study into moist plumes related to overshooting convection in the Asian monsoon anticyclone (AMA). The primary data source for their analysis is from the EU StratoClim flight campaign in Summer 2017, and they supplement this well with satellite observations and an ensemble trajectory analysis. The paper is well-written, informative, and enjoyable to read, and their findings regarding the persistence of moisture in the AMA are new and unique. I have some minor concerns related to the trajectory analysis utilized in this study that are addressed in the comments below, as well as some suggestions to improve readability of the paper, particularly regarding the figures.

General Comments:

1. There should be some more discussion of the limitations associated with long-term trajectory analyses, especially when using 30-day backwards trajectories. While using ERA-5 with a higher temporal resolution may improve the reliability of the trajectories compared to previous generation reanalyses, there is still a concern of large spatial errors given the length of some of the trajectory-matched convective sources presented in Figure 5c.

2. The figures presented throughout the paper are very informative but are also quite busy and hard to digest. One major improvement that I suggest would be to find a color table that is less harsh than rainbow for many of the Figures. I think rainbow is fine for designating individual flights like in Figure 5c, but when it comes to analysis in Figure 1, Figure 5a and b, etc., a less harsh color table will make the analysis feel less overwhelming on a first time read through. This will also allow layered contours of different variables to stand out against the background color.

Specific Comments:

1. Line 36 – A citation should be added for further information on stratospheric ozone chemistry, specifically Anderson et al. 2017 and similar papers discuss the potential for ozone destruction via activation of inorganic chlorine due to convective increases of H2O in cold lower stratosphere environments
2. Line 79 – Please define FP7
3. Lines 152-155 – The horizontal resolution and some precision/accuracy information about the MLS measurement should also be provided here
4. Lines 158-163 – As above, please provide a sentence regarding horizontal resolution and precision/accuracy information

5. Lines 176-178 – This sentence should either be reworded or include an addition to provide a little more detail into the ozone-water vapor relationship and how that is a reliable method for diagnosing the transition layer
6. Line 184-185 – This sentence should include a citation
7. Lines 184-188 – Some discussion of above anvil cirrus plumes could be warranted in this paragraph, see Homeyer et al. 2017 (https://doi.org/10.1175/JAS-D-16-0269.1 ) and O'Neill et al. 2021 (https://www.science.org/doi/10.1126/science.abh3857 )
8. Lines 190-196 – As noted in general comments, some discussion of trajectory errors and limitations is appropriate here
9. Line 211 – What is the reasoning for examining convective hits at 100 hPa only? Could this have an impact on your results where 100 hPa is farther into the stratosphere in the northern AMA (with high tropopause pressures) and closer to the tropopause in the southern part of the region?
10. Lines 213-214 – A sentence further describing this method that is shown in Bucci et al. could be helpful, especially to indicate how convective hits would not be overly dependent on parcel age at time of convection due to trajectory ensemble spread over time.
11. Lines 219-221 – This sentence is a bit wordy, I would suggest rewording and breaking into two separate sentences
12. Figure 1 – This figure has a lot of important information, but it is a bit overwhelming and hard to interpret. Following my general suggestions above, I think that reworking the color table for this figure and being sure to prominently display overlayed contours could help. Additionally, the black pixels representing likely sources of hydrated features seems important and should be emphasized; the black pixels almost make it look like a region of missing data on a first read through.
13. Figure 1c and d–The vertical dashed lines could be labeled as they are in Figure 5 for consistency throughout the figures, and it could also help to clearly indicate what they are
14. Lines 236-237 – For clarity, this should include a citation and some further explanation
15. Line 240 – I think it would be helpful here to specify the time by which CLS water vapor mostly returns to late July values
16. Line 269 – What is the bin size here?
17. Lines 290-291 – Citation needed here
18. Figure 5 – There is a lot going on in this figure as well, a couple of suggestions: (1) it would be helpful for the flight numbers to be as bold (as they are in the legend). (2) making the circles indicating the location of sampling more prominent by filling them in, making them larger, or both. (3) Increasing the density of wind vectors could help to fill in the areas of the map that are fairly empty, and also will help draw attention to them (it took me a while to notice them at first).
19. Line 310/Figure 5 – Given that C2 represents a different observation than A2 and B2 for flight F2, it may be helpful to visually indicate that, perhaps with a 'C2*'. When first reading through, I was unsure if C2 was mislabeled until it was first mentioned on page 11.

20. Lines 347-348 – What are these measurements coming from? Flight data? MLS? And is this the mean for the entire domain? Is it the mean for the entire warm/wet period, or just a selected portion?
21. Line 372-373 – How many of the trajectories intersected the convective system vs. passing just south of it?
22. Lines 402-403 – I suggest that this statement should be a little stronger, something along the lines of "It is still possible that some of these…" rather than "this does not rule out that…"
23. Line 450 – Suggest changing 'a lot of evidence for' to 'ample sampling of'
24. Line 483-485 – Suggest breaking this sentence into two separate sentences
25. Page 14 – It would be nice to include a sort of 'looking forward' paragraph to the conclusions, particularly with regards to the future of *in situ* observations of stratospheric H2O. One example would be the ongoing Dynamics and Chemistry of the Summer Stratosphere (DCOTSS) field campaign, which should help to provide similar observations in the North American Monsoon Anticyclone.

Technical Corrections:

1. Line 21 – 'the key contributor' should be changed to 'a key contributor'
2. Lines 120-121 – FISH instrument should be changed to "The FISH instrument" and "in flight" should be changed to "for flights"
3. Line 221 – typo: 'sown' instead of 'shown'
4. Line 253 – 'are hardly' could be replaced by 'cannot'
5. Figure 5 – There are two flights labeled F4, and no flight labeled F5.
6. Line 350 – 'Fig. 6a,b' could be replaced by 'Figs. 6a and 6b'
7. Line 396 – One hundred? Or multiple hundreds?
8. Line 474 – 'that the convective' should be changed to 'that convective'

---

## Author Comment (AC1)

**Response to the Anonymous Referee #1.**

**We express our sincere gratitude to Anonymous Referee #1 for constructive remarks on the manuscript and the appreciation of our study.**

*I have three general, rather minor comments that I would like to make. First, even though the focus of this study is the Asian summer monsoon, it is important to remind us that the North American monsoon also plays role in the lower stratospheric water vapor budget. There are some parts of the text that needs to be clarified (see the first specific comment below).*

**We agree that the North American monsoon may be as important in terms the convective transport into the stratosphere as its Asian counterpart. The respective modifications (following specific remarks) have been implemented into the manuscript.**

*Second, I would like to see some statements about uncertainties in the trajectory model results. Can we trust 100% of the result? How sensitive is the result to the input variables or dynamical fields?*

**The following text has been added into Sect. 2.4.:**

> **"It should be noted that the trajectory model integrates 1000 backward trajectories per data point along the flight track which are submitted to a random noise equivalent to a diffusion D=0.1 $m^2 s^{-1}$ as in Bucci et al. (2020). As such, the integration is a discretization of the adjoint equation of the advective diffusive equation, which is well posed for backward integration (Legras et al., 2005). Unlike single-trajectory Lagrangian calculations, this method does not generate spurious small-scale features as backward time increases, and can be shown to converge with time for a pure passive scalar. With that, the trajectories from each data point come from several, possibly, many sources and the results presented are a statistics over these 1000 trajectories.**
>
> **Obviously, the results can be affected by biases in the wind field, heating rates and the cloud height product used in this study. The ERA5 is presently considered as the most advanced reanalysis and it was shown to display very consistent transport properties of diabatic versus kinematic trajectories (Legras & Bucci, 2020), which are in excellent agreement with observations (Brunamonti et al., 2018; von Hobe et al., 2021). The main concern in the Asian monsoon region is that ERA5 displays high penetrative convection over the Tibetan Plateau, which might bias the heating rates in the upper TTL over this region (SPARC S-RIP report, 2021). As the trajectories involved in this studied are mostly outside the Plateau, we do not expect any significant impact."**

*Third, the figures in the manuscript and the supplement material are outstanding and yet complex. Some figures are probably too complex. I had hard time understanding Figure 1, in particular, as it contains so many variables and colors. It would be helpful to revisit the figures and improve if there is time.*

**The figures have been modified following the specific remarks as detailed below.**

*P1, L21 – 'the wettest region' can be replaced by 'one of the wettest regions'*

*P1, L30 – 'water vapour enhancements' can be replaced by 'water vapour measurements during the campaign'*

*P2, L40 – 'the primary contributor' could be replace by 'one of the primary contributors'*

*P3, L82 – 'in the other flights' could be replaced by 'the rest of the flights'*

*P4, L136-137 – Here, 'distribution' could be replaced by 'distributions' on both sentences.*

**All suggestions implemented.**

*P2, L57-60 – It would be helpful to add a sentence summarizing the findings, e.g., is the degree of convective impact different? Or do the source regions differ in different studies?*

**The following sentence has been added: "In general, there is no consensus regarding the primary convective source regions, nor regarding the net convective effect of deep convection on the CLS water vapour, which points out the complexity of physical processes in the AMA system."**

*P3, L86-87 – Do Stroh et al. (2021) include the description of the instruments as well?*

**Stroh et al. (in prep. 2022) include a brief description of the instruments summarized in a table.**

*P3, L107 – Does 'under UTLS conditions' mean low temperature and low humidity?*

**Yes, the respective sentence has been modified: "…under UTLS conditions, i.e. low temperature and humidity environment."**

*P4, L126 – Here, 'They' refers to Singer et al. (2021)?*

**Yes, the sentence has been modified respectively.**

*P5, L153 – The meaning of 'evenly distributed' is unclear.*

**The phrase "evenly distributed" has been removed.**

*Also, there is a newer version of MLS H2O (v5), which became available more than a year ago. It is also known that MLS H2O (v4.2) has drift issues in the stratosphere.*

**The satellite data analysis for this study was performed before the MLS v5 retrieval has been issued. We are aware of the drift issues in v4.2 data version however this is of little relevance for this study since we use MLS H2O data mostly to describe the regional and subseasonal variability in the AMA region. A quick comparison of the results presented in Fig. 1 using v4.2 and v5 retrievals did not show any significant differences.**

*P5, section 2.3 – I think this section provides useful information. It would be helpful to add a sentence explaining the purpose of this section here.*

**The following sentence was added in the beginning of the section: "In this section, we define the key terms regarding the vertical structure o AMA and physical processes therein."**

*P5, L178 – It is unclear why the thermal approach to TTL definition is only suitable for this study. Is this related to the fact that water vapor is sensitive to vertical structure of temperature?*

**The thermal approach to TTL definition is suitable for this study because it relies on the local measurements as those exploited in this study. The text has been reformulated to avoid confusion.**

*P5, L185-186 – It would be useful to include references for the hydration vs. dehydration processes in the TTL.*
**References to Jensen et al., 2007 and Schoeberl et al., 2018 have been added here.**

P6, L197 – Are there any references for HIMAWARI-8 could be cited here?

   **The following reference has been added:**

   Bessho, K., Date, K., Hayashi, M., Ikeda, A., Imai, T., Inoue, H., Kumagai, Y., Miyakawa, T., Murata, H., Ohno, T., et al.: An introduction to Himawari-8/9—Japan's new-generation geostationary meteorological satellites, Journal of the Meteorological Society of Japan. Ser. II, 94, 151–183, https://doi.org/https://doi.org/10.2151/jmsj.2016-009, 2016

*P6, L207 – Would 'a specific version of product based on the version 2018.1' be the same as 'version 2018.1'?*

**Unnecessary or redundant information has been removed from the text**

*P6, L211 – Is '100 hPa' an arbitrary threshold or based on a statistical analysis?*

**The 100 hPa threshold was set mostly arbitrarily. The statistical analysis shows a rapid decrease of the number of convective hits above 100 hPa. The lower-pressure threshold does not allow identifying convective sources for the lower-altitude hydration features (e.g. A2), whereas the higher-pressure threshold adds ambiguity to the source identification for higher-altitude features (e.g. B7).**

*P6, L219 – How is the 2017 Asian monsoon season characterized as a stable anticyclone? Compared to 10-year climatology or compared to previous three years? Some statistics might be helpful here.*

**The text has been modified: "The 2017 Asian monsoon season was not marked by an anomalous dynamical behavior (Manney et al., 2021), however the campaign occurred during a break - active transition. The strongest convective activity took place above the Southern slopes of Himalayas and the Tibetan plateau during late July and early August as can be inferred from…"**

*Fig. 1a - It is not easy to tell from Fig. 1a. In Fig. 1a, the water vapor contours can be smoothed (1-2-1 smoothing) or one can use bigger grid boxes to show smooth contours. Also, what do black colors mean in Fig. 1a? I think the wind vectors can be improved here as well (likewise in Fig. 5). For instance, one can put wind vectors every 2.5 degree latitude instead of 5.*

*Fig. 1b – I think the AMA boundary looks rather too broad here. Adding more contours or choose a smaller threshold of Montgomery stream function might work better*

**All remarks on Fig. 1 have been implemented**

*P7, L241 – Is Brunamonti et al. (2018) also based on summer of 2017?*

**Yes, we have included the respective mention in the text.**

*P7, L247 – I would recommend using a quantitative adjective than 'striking' here, e.g., 'large' variability.*

**Replaced by "remarkable".**

*P7, L261 – Do 'those' refer to the high RHi values?*

**'Those' refer to the subsaturated cloud occurrences. The text has been rephrased: "Such occurrences are mainly caused by…"**

*P7, L263 – It would be helpful to add an explanation about 'Lagrangian temperature history' here.*

**The sentence has been modified: "…depends on the air parcel's (Lagrangian) temperature history".**

*P7, L274- Is '14%' higher or lower than any statistics or expectations?*

**The following sentence has been added: "This is consistent with a comprehensive analysis of airborne data from various campaigns by Kraemer et al., 2020, who pointed out a significantly larger amounts of IWC in subsaturated ice crystals above the CPT in AMA compared to that in the surrounding tropical regions, thereby upholding the importance of AMA as the source of LS water."**

*Comments for Fig. 4 It would be useful to add approximate altitude for Figs. 4a and 4b. Also, in Fig. 4b, high delta D exists as high as 420K potential temperature surface.*

**Approximate altitude added. The high deltaD values are not totally surprising at this level given the observed hydrated features up to 415 K and their heating rate of more than 1 K/day.**

*P9, Section 4.2 – I am wondering is there a way to quantify the uncertainty in the derived convective age. Is it sensitive to the type of cloud top data and also meteorological fields?*

**The following text has been added: "We note that while the 1σ-error of the age estimates is generally less than an hour, the attribution of convective sources largely depends on the cloud top data. In particular, the improved v2018.1 trajectory product coupled with NWF SAF geostationary data analysis provided a qualitatively better correlation between the distribution of convective hits and wet-and-heavy parcels as compared with the product used by Bucci et al. 2020.."**

*P10, L340 – Does 'mixing ratio enhancement' mean the actual water vapor mixing ratios or only the increased amount of water vapor?*

**It means water vapour mixing ratio enhancement, as specified now in the respective sentence**

*P10, L350 – Fig. 6a,b could be replaced by Figs. 6a and 6b*

**Done.**

*P10, L350 – Does 'at this level' mean local CPT?*

**Yes, sentence modified.**

*P10, L359 – Here, 'such an amount' could be replaced by 'such high amount'. I am also wondering what is the mechanism that enables the convective plume preserve high water vapor for 5 days.*

**Replacement done. A convective plume that doesn't experience supersaturation can preserve enhanced vapour in the CLS for days and perhaps weeks, provided the weaker mixing at these levels compared to the upper-TTL.**

*P11, L373 – 'intersecting a large convective system' – It looks like the trajectories lie between the large convective system and the group of small cells to the west.*

**The locations of convective hits (i.e. where the traced air parcels hit the convective cloud) are indicated in Fig.7a as black-filled red circles. A clarification has been added to the text.**

*Comments for Fig. 7c I assume temperature means potential temperature in Fig. 7c. Also, it is hard to locate 140 or 160% RHi in this figure as explained in the text.*

**Figure 7 has been reworked to display the RHi evolution versus the y-axis.**

P11, L389 – Does 'across' mean from below to above the CPT?

**Yes, the text has been modified accordingly.**

P11, L393 – It would be helpful to give the time marks for the presence of subvisible cirrus clouds in Fig. 8a.

**There are six occurrences of cirrus clouds during this flight segment. They are tagged and color-coded along the potential temperature time series.**

*Comments for Fig. 9a Fonts for T1-B7 could be bigger.*

**Done.**

*P12, L435 – Instead of 'around the CPT', near or close to might be more accurate.*

**Changed to "near CPT level"**

*P13, L448 – Does 'there is typically no more than one case' apply to all the cases referred above?*

**The sentence has been modified: "…there is typically no more than one case of water vapour enhancement above the tropopause detected during a given field campaign."**

*P13, L454 – In this sentence, the meanings of 'over these regions' and 'in the summer monsoon anticyclones' are not clear. Are those referring to the North American monsoons?*

**The sentence has been modified: "..support the role of overshooting convection in maintaining the water vapour maximum in the North American and Asian monsoon anticyclones."**

*P13, L467 – 'around the tropopause' could be replaced by 'near the tropopause'.*

*P14, L483 – This sentence could be split into two -> introduction. However,…*

**All done.**

*P14, L505 – It would be helpful to add a sentence about what the authors think the future direction or need is in terms of field studies related to StratoClim.*

The following paragraph has been added: "Further insights into the AMA gaseous/particular composition and dynamics will be provided by an upcoming airborne campaign within the Asian summer monsoon Chemical and Climate Impact Project (ACCLIP; https://www2.acom.ucar.edu/acclip), which will sample the Western Pacific mode of the monsoon and eastward eddy shedding using NASA WB-57 and NCAR GV aircrafts. The stratospheric impact of overshooting convection in the North American monsoon is a primary target of the Dynamics and Chemistry of the Summer Stratosphere (https://dcotss.org/) project, involving ER-2 high -altitude aircraft.

---

## Author Comment (AC2)

**Response to the Anonymous Referee #2.**

We thank the Anonymous Referee #2 for constructive remarks on the manuscript.

*General Comments:*

1. *There should be some more discussion of the limitations associated with long-term trajectory analyses, especially when using 30-day backwards trajectories. While using ERA-5 with a higher temporal resolution may improve the reliability of the trajectories compared to previous generation reanalyses, there is still a concern of large spatial errors given the length of some of the trajectory-matched convective sources presented in Figure 5c.*

   **The following text has been added into Sect. 2.4.:**
   **"It should be noted that the trajectory model integrates 1000 backward trajectories per data point along the flight track which are submitted to a random noise equivalent to a diffusion D=0.1 m2 s−1 as in Bucci et al. (2020). As such, the integration is a discretization of the adjoint equation of the advective diffusive equation, which is well posed for backward integration (Legras et al., 2005). Unlike single-trajectory Lagrangian calculations, this method does not generate spurious small-scale features as backward time increases, and can be shown to converge with time for a pure passive scalar. With that, the trajectories from each data point come from several, possibly, many sources and the results presented are a statistics over these 1000 trajectories.**
   **Obviously, the results can be affected by biases in the wind field, heating rates and the cloud height product used in this study. The ERA5 is presently considered as the most advanced reanalysis and it was shown to display very consistent transport properties of diabatic versus kinematic trajectories (Legras & Bucci, 2020), which are in excellent agreement with observations (Brunamonti et al., 2018; von Hobe et al., 2021). The main concern in the Asian monsoon region is that ERA5 displays high penetrative convection over the Tibetan Plateau, which might bias the heating rates in the upper TTL over this region (SPARC S-RIP report, 2021). As the trajectories involved in this studied are mostly outside the Plateau, we do not expect any significant impact."**

2. *The figures presented throughout the paper are very informative but are also quite busy and hard to digest. One major improvement that I suggest would be to find a color table that is less harsh than rainbow for many of the Figures. I think rainbow is fine for designating individual flights like in Figure 5c, but when it comes to analysis in Figure 1, Figure 5a and b, etc., a less harsh color table will make the analysis feel less overwhelming on a first time read through. This will also allow layered contours of different variables to stand out against the background color.*

   **The color table in Fig. 1 has been changed to a less harsh one. Some modifications have been applied to other figures to improve their readability.**

   *Line 36 – A citation should be added for further information on stratospheric ozone chemistry, specifically Anderson et al. 2017 and similar papers discuss the potential for ozone destruction via activation of inorganic chlorine due to convective increases of H2O in cold lower stratosphere environments*

   **Obviously, Anderson et al., 2017 is very relevant for this study. The citation has been added.**

*Line 79 – Please define FP7*

**Done.**

*Lines 152-155 – The horizontal resolution and some precision/accuracy information about the MLS measurement should also be provided here*

**Done.**

*Lines 158-163 – As above, please provide a sentence regarding horizontal resolution and precision/accuracy information*

**Done.**

*Lines 176-178 – This sentence should either be reworded or include an addition to provide a little more detail into the ozone-water vapor relationship and how that is a reliable method for diagnosing the transition layer.*

**The text has been modified: "Another approach is based on the TTL thermal structure, where the lower and upper boundaries are defined respectively as the level of minimum stability and the cold point tropopause (CPT) (Gettelman and Forster, 2002). Pan et al. (2014) found that the thermally-defined TTL boundaries are consistent with those derived from the ozone-water vapour relationship. In this study, we adopt the thermal definition of the TTL as in this case the boundaries can be derived from the local instantaneous measurements provided by the Geophysica aircraft"**

*Line 184-185 – This sentence should include a citation*

**Reference to Danielsen, 1993 has been added.**

*Lines 184-188 – Some discussion of above anvil cirrus plumes could be warranted in this paragraph, see Homeyer et al. 2017 (https://doi.org/10.1175/JAS-D-16-0269.1 ) and O'Neill et al. 2021 (https://www.science.org/doi/10.1126/science.abh3857 )*

**The paragraph has been modified: "A convective overshoot (also termed "ice geyser" by Khaykin et al. (2009)) is defined as detrainment of ice crystals above the local CPT (Danielsen, 1993). Depending on the relative humidity at the level of detrainment, this process can lead either to CLS moistening by rapid ice sublimation, or to irreversible dehydration via uptake of vapour by the injected ice crystals, their growth and sedimentation (e.g. Jensen et al., 2007; Schoeberl et al., 2018). The clouds that have formed in the CLS as a result of local cooling are termed *in situ* cirrus. An *in situ* cirrus cloud is not to be confused with the above anvil cirrus plume (AACP), which is a plume of ice and water vapour in the LS that occurs in the lee of overshooting convection (Homeyer et al., 2017; O'Neill et al., 2021). A *secondary cloud* refers to an *in situ* cirrus that has nucleated from an air mass enhanced in water vapour as a result of convective overshoot."**

*Lines 190-196 – As noted in general comments, some discussion of trajectory errors and limitations is appropriate here.*

**Cf. response to general remarks above.**

*Line 211 – What is the reasoning for examining convective hits at 100 hPa only? Could this have an impact on your results where 100 hPa is farther into the stratosphere in the northern AMA (with high tropopause pressures) and closer to the tropopause in the southern part of the region?*

**The statistical analysis of convective hits shows a rapid decrease of the number of convective hits starting above 100 hPa. While the lower-pressure threshold does not allow identifying convective sources for the lower-altitude hydration features (e.g. A2), the higher-pressure threshold adds ambiguity to the source identification for the higher-altitude and/or older-age features (e.g. B7). As for the higher tropopause pressure in the northern AMA, this should have very limited impact on the analysis since the warmer temperatures at and above the tropopause in this region enable high water vapour mixing ratios (cf. Fig. 1a). Otherwise said, the altitude of convective detrainment relative to the local tropopause in the subtropical AMA is much less important than in the Southern AMA, where the colder tropopause limits the hydration potential of overshooting convection.**

*Lines 213-214 – A sentence further describing this method that is shown in Bucci et al. could be helpful, especially to indicate how convective hits would not be overly dependent on parcel age at time of convection due to trajectory ensemble spread over time*

**The respective text in Sect. 2.4 has been modified and completed (cf. response to general remarks above).**

*Lines 219-221 – This sentence is a bit wordy, I would suggest rewording and breaking into two separate sentences*

**The text has been modified: "The 2017 Asian monsoon season was not marked by an anomalous dynamical behavior (Manney et al., 2021), however the campaign occurred during a break - active transition. The strongest convective activity took place in late July and early August above the Southern slopes of Himalayas and the Tibetan plateau as shown by…"**

*Figure 1 – This figure has a lot of important information, but it is a bit overwhelming and hard to interpret. Following my general suggestions above, I think that reworking the color table for this figure and being sure to prominently display overlayed contours could help. Additionally, the black pixels representing likely sources of hydrated features seems important and should be emphasized; the black pixels almost make it look like a region of missing data on a first read through. Figure 1c and d–The vertical dashed lines could be labeled as they are in Figure 5 for consistency throughout the figures, and it could also help to clearly indicate what they are*

**The color table in Fig. 1 has been changed to appear less harsh and to improve the readability of the overlaid contours. The pixels, representing the likely sources of hydrated features are now colored pink. The vertical dashed lines in Fig. 1c and d have been labeled.**

*Lines 236-237 – For clarity, this should include a citation and some further explanation*

**The text has been modified: "This "cold/dry" period is marked by stronger convective activity in the region reflected by low OLR (Fig. 1c), i.e. colder and higher cloud tops, and higher carbon monoxide mixing ratio (Fig. 1d). Since the carbon monoxide is a tracer for troposphere to stratosphere transport,**

**the elevated CO concentration in the LS is indicative of the enhanced upward flux across the tropopause (e.g. Randel et al., 2010.).”**

*Line 240 – I think it would be helpful here to specify the time by which CLS water vapor mostly returns to late July values*

**Added “…by mid-August”.**

*Line 269 – What is the bin size here?*

**Bin size is 0.1 ppmv by 1 K (added to Fig.3 and Fig. 4 captions).**

*Lines 290-291 – Citation needed here*

**References to Moyer et al., 1996; Hanisco et al., 2007 have been added here.**

*Figure 5 – There is a lot going on in this figure as well, a couple of suggestions: (1) it would be helpful for the flight numbers to be as bold (as they are in the legend). (2) making the circles indicating the location of sampling more prominent by filling them in, making them larger, or both. (3) Increasing the density of wind vectors could help to fill in the areas of the map that are fairly empty, and also will help draw attention to them (it took me a while to notice them at first).*

**All remarks on Fig. 5 have been implemented.**

*Line 310/Figure 5 – Given that C2 represents a different observation than A2 and B2 for flight F2, it may be helpful to visually indicate that, perhaps with a 'C2*'. When first reading through, I was unsure if C2 was mislabeled until it was first mentioned on page 11.*

**Done.**

*Lines 347-348 – What are these measurements coming from? Flight data? MLS? And is this the mean for the entire domain? Is it the mean for the entire warm/wet period, or just a selected portion?*

**This is based on the airborne measurements; a mention has been added to the text.**

Line 372-373 – How many of the trajectories intersected the convective system vs. passing just south of it?

**The following sentence has been added: “The fraction of trajectories intersecting this convective system amounts to 47%.”**

Lines 402-403 – I suggest that this statement should be a little stronger, something along the lines of “It is still possible that some of these…” rather than “this does not rule out that…”

**The text has been modified: “It is however still possible that some of these crystals were produced by overshooting”.**

Line 450 – Suggest changing 'a lot of evidence for' to 'ample sampling of'

**Done.**

Line 483-485 – Suggest breaking this sentence into two separate sentences

**Done.**

Page 14 – It would be nice to include a sort of 'looking forward' paragraph to the conclusions, particularly with regards to the future of in situ observations of stratospheric H2O. One example would be the ongoing Dynamics and Chemistry of the Summer Stratosphere (DCOTSS) field campaign, which should help to provide similar observations in the North American Monsoon Anticyclone.

**The following paragraph has been added: "Further insights into the AMA gaseous/particular composition and dynamics will be provided by an upcoming airborne campaign within the Asian summer monsoon Chemical and Climate Impact Project (ACCLIP; https://www2.acom.ucar.edu/acclip), which will sample the Western Pacific mode of the monsoon and eastward eddy shedding using NASA WB-57 and NCAR GV aircrafts. The stratospheric impact of overshooting convection in the North American monsoon is a primary target of the Dynamics and Chemistry of the Summer Stratosphere (https://dcotss.org/) project, involving ER-2 high -altitude aircraft.**

*Technical Corrections:*

*1. Line 21 – 'the key contributor' should be changed to 'a key contributor'*
*2. Lines 120-121 – FISH instrument should be changed to "The FISH instrument" and "in flight" should be changed to "for flights"*
*3. Line 221 – typo: 'sown' instead of 'shown'*
*4. Line 253 – 'are hardly' could be replaced by 'cannot'*
*5. Figure 5 – There are two flights labeled F4, and no flight labeled F5.*
*6. Line 350 – 'Fig. 6a,b' could be replaced by 'Figs. 6a and 6b'*
*7. Line 396 – One hundred? Or multiple hundreds?*
*8. Line 474 – 'that the convective' should be changed to 'that convective'*

**All done.**

---

## Author Comment (AC3)

**Response to the Anonymous Referee #3.**

**We express our sincere gratitude to Anonymous Referee #3 for the positive review and useful remarks on the manuscript.**

*L291 Please provide a reference in support of this statement.*

**References to Moyer et al., 1996; Hanisco et al., 2007 have been added here**

*L303 What number of convective clouds are identified in each case? It is too hard for the reader to integrate by eye the proportion of convective hits over the wet parcels for each flight. e.g. From Fig S3, F1, F5 and F6 appear to have very few wet parcels exceeding 1 standard deviation. I am surprised they appear to have identified as many convective cloud intercepts as other flights. Also, I presume F8 isn't shown in Fig 5 for this reason (no wet parcels along the flight track) but it should be explained somewhere.*

**As a matter of fact, nearly all the parcels sampled in the CLS have convective encounters along the 30-day trajectories. The trajectories from each data point come from several, possibly many, sources and the results presented are a statistics over these 1000 trajectories. This way, the fraction of convective hits for each 1-second sample provides a quantitative estimate of the likelihood of a parcel to originate from a convective cloud. What is evidenced by Fig. 4, 6 and S3 is that the distribution of wet-and-heavy parcels in the theta-H2O space is qualitatively correlated with the distribution of parcels originating from deep convection (represented by high convective hits fraction).**

**The fraction of wet-and-heavy parcels above the CPT varies from one flight to another. We have added the following text into the first paragraph of Sect. 4.2: "The fraction of wet-and-heavy parcels to all parcels sampled above the local CPT varies from 0.5% (F6) to 11% (F2) between the different flights. No wet-and-heavy parcels have been detected in F8, which is why it is not displayed in Fig. 5"**

*L310and L317 Reading is made harder by occasional forward references in the early results sections, particularly L310 and L317. Please reconsider the use of forward references. I would suggest mentioning the convention of labelling hydration events when you point to Supplementary Fig. S4 near L306, rather than referencing forward, since that is where the reader is currently expected to look if they want to see individual moist features.*

**The hydrated feature labelling convention has been re-introduced into Sect. 4: "The results for the individual flights are provided in Fig. S3 of the Supplement. The hydrated features (layers of enhanced water vapour, exceeding 1-σ of the campaign ensemble) are denoted throughout the article as _Ax_ or _Bx_, where _x_ is the flight number.**

*L330 This sentence is hard to follow. Can the text please specify the flight paths being referred to? Does it mean the paths taken by F1-F7? Or is it referring to F1,F2,and F4 being over Nepal? Perhaps I misunderstand, but F3 seems an obvious exception to the flights F1-4 during the warm/wet period. I am not sure that the argument about intercepts in space can be separated from the impact of time differences.*

**The sentence refers to all the flight shown in Fig.5 and the message here is that the wet-and-heavy parcels sampled along the flight tracks are mostly restricted to the northern part of the flight domain,**

**i.e. north of 25 N. The only exception is F7, in which a hydration feature was detected at 22-23 N. Some clarifications have been added to the text.**

*L343 I found the shift of terminology from 'period' to 'regime' unclear. This is first use of the word regime even though it refers back to section 3 where 'period' is used (and section 3 heading uses 'mode'). Suggest rephrase introduction of the word regime here, or a more consistent terminology introduced in Section 3.*

**The term "mode" is now used throughout the manuscript.**

*L364 A2 must also be displaced horizontally, which is not shown in Fig 6. How extensive is that? Is it conceivable that such a body of air remained intact after 5 days? You could test this by calculating a series of back trajectories released from vertical and horizontal positions between A2 and B2. Or perhaps more precisely, forward trajectories, allowing for ice settling, from the suspected original convective event for B2. This is a point of interest, I do not expect any revision for this.*

**A2 is indeed displaced horizontally. Given the aircraft cruising speed of 180 m/s, the distance between the A2 midpoint and B2 peak H2O mixing ratio, as can be inferred from Fig. 6c, is about 250 km. However, since A2 and B2 are separated by 500 m in altitude (or 9 K in potential temperature), their back trajectories are substantially different: those for A2 lie closer to the AMA center.**

*L467 It would be of general help to the reader to provide the summary with more backward references to specific results. In particular at L467, a reference to the analysis supporting this conclusion of a drastic drop of water vapour. It might be interpreted that all of Section 1 is relevant, but Fig 1 seems to be the key result for this sentence. It's also important so that a reader does not mix up this convectively-modulated temperature effect from convective lofting processes.*

**The paragraph has been reworked: "The warm/wet mode sampled during the early flights revealed substantial enhancements of water vapour mixing ratio reaching above 10 ppmv (twice the background) as high as 400 K (18.2 km) level, but very little evidence for dehydration upstream. By contrast, the second (cold/dry mode) period of the campaign with organized large-scale convection inside and close to the flight domain led to synoptic-scale CPT cooling and a drastic drop of water vapour by ~30% near the tropopause. We note though that the dehydration layer did not extend above 395 K, whereas in the upper layers, the excess of water vapour was subject to a transient phase transition, resulting in an outbreak of cirrus at levels up to 415 K (18.9 km). A similar inference was made by Brunamonti et al. (2018) on the basis of balloon soundings of water vapour and ozone in Nepal as part of StratoClim campaign in 2017. They argued that overshooting convection is responsible for an isolated maximum of H2O in the CLS observed in July 2017, whereas the water vapour minimum at the CPT level is caused by synoptic-scale cold anomaly above the southern slopes that maximized around 9 August."**

Technical corrections: L221 typo 'shown' L335 'at' seems unnecessary. Remove? L367 The bibliography is missing Kim and Alexander 2015. A quick glance finds Muller and Peter 1992 also missing. Please double-check your bibliography is complete. Fig1 Figs 1c and 1d are in the reverse order compared to the text in the caption and main body. Correct the text or the figure order. Fig5 F5 appears to be incorrectly labelled as F4. Please correct. FigS4 The hydrated feature for flight 5 is marked A7. I think it is meant to be A5?

**All done.**

Fig5 Refers to a hydrated feature C2 but is not mentioned. Please clarify. Is this associated with the ice cloud identified in Fig 6? Or is it meant to be hydrated feature A3 (as the colouring suggests)?

**C2 (now referred to as C2\*) is associated with the ice cloud identified in Fig. 6. The annotation marks the location of the source convection, which happens to coincide with that for A3.**

---

## Editor Decision (ED1)

**Persistence of moist plumes from overshooting convection in the Asian monsoon anticyclone**

Sergey M. Khaykin1, Elizabeth Moyer2, Martina Krämer3, Benjamin Clouser2, Silvia Bucci4,a, Bernard Legras4, Alexey Lykov5, Armin Afchine3, Francesco Cairo6, Ivan Formanyuk5, Valentin Mitev7, Renaud Matthey8, Christian Rolf3, Clare Singer2,b, Nicole Spelten3, Vasiliy Volkov5, Vladimir Yushkov5 and Fred Stroh3

[revised manuscript text omitted]

---

## Author Response (AR2)

This is a tracked-changes document with comments to respond the remarks from the Editor.

[revised manuscript text omitted]

a)

[Figure]

b)

[Figure]

c)

[Figure]

RHi history of the secondary cloud in F2 (C2*)

**Figure 7.** Trajectory analysis of the secondary cloud (C2) detected in F2 (29 July). (a) Back trajectories (green curves) and convective hits locations (black-filled red circles) superimposed onto the IR BT at the time of convective hits (cf. time stamp in the panel). (b) Same as (a) but for the time of the MLS sampling of the moist plume. The spatiotemporally-collocated MLS swath is displayed as markers color-coded by H2O mixing ratio at 393 level. The wettest MLS measurement coincides with the location of hydrated parcels. (c) Relative humidity over ice evolution along the back trajectories with color coding by potential temperature computed from ERA5 temperature and assumed H2O mixing ratio of 12 ppmv (see text for details). The balck markers show the locations of convective hits, the vertical dashed line indicate the time of MLS sampling of the moist plume on 24 July.

a)

[Figure]

F7, water vapour, total water, saturation mix. ratio, potential temperature, scattering ratio

b)

[Figure]

c) F7, water vapour and isotopic ratio

[Figure]

Figure 8. Results of F7 measurements (08 August) with the features of interest marked A7 and B7. (a) Time series of potential temperature θ (left-hand axis) and water vapour/total water (right-hand axis). The θ time series is tagged by cloud occurrence and color-coded by scattering ratio. The IWC is shown as cyan shading stacked on the water vapour curve, the darker cyan curve depicts the total water (right-hand axis). The dashed magenta line depicts the saturation mixing ratio (right-hand axis). The CPT level displayed as black dashed line is inferred from airborne measurements. (b) Vertical profiles of water vapour (black circles), total water (cyan) and IWC (cyan shading). The red dashed curve indicates the mean water vapour profile form the previous flights (F1 - F6). (c) Binned water vapour profile with pixels color-coded by HDO/H2O isotopic ratio. The black solid curve depicts the campaign-median H2O profile, the dotted curves represent two standard deviations respectively.

[Figure]

[Figure]

1150

1155

**Figure 9. Evolution of cloud occurrence and RHi along the backtrajectories released from B7 (H2O enhancement at 410 K in F7). (a) Backtrajectories (black curves) and ground tracks of CALIPSO and CATS satellite lidars color coded by the cloud top altitude. The magenta circles mark the locations of the tracked parcels at the time of the nearest satellite transacts. The red labels indicate the transacts number (T1 – T6) and its UTC. The arrows are wind vectors (ERA5) interpolated at 410 K level. The track of F7 flight is shown as solid curve with altitude color-coding. (b) Attenuated scattering ratio (SR) for T3 (contours, 3 and 6 SR units) and T4 (color map). The white oval shows the latitude-altitude location of the B7 parcels at the time of T4 transect. The grey dashed curve is the projection of F7 flight track onto the lidar section. Note that the location of B7 parcels matches the evaporating part of the cloud. (c) Evolution of RHi along the backtrajectories computed using ERA5 temperatures and measured $H_2O$ mixing ratio in B7 (7.3 ppmv). The shading indicates two standard deviations, the vertical lines with red labels mark the timing of satellite transacts, the red dashed line shows homogeneous freezing threshold for the given temperature and humidity.**